# Enhancing Reasoning for Diffusion LLMs via Distribution Matching Policy Optimization

Yuchen Zhu [* 1]  Wei Guo [* 1]  Jaemoo Choi [1]  Petr Molodyk [1]  Bo Yuan [1]  Molei Tao [† 1]  Yongxin Chen [† 1]

## Abstract

Diffusion large language models (dLLMs) are promising alternatives to autoregressive large language models (AR-LLMs), as they potentially allow higher inference throughput. Reinforcement learning (RL) is crucial to enabling dLLMs to achieve performance comparable to that of AR-LLMs on important tasks, such as reasoning. However, RL algorithms well-suited to dLLMs' unique characteristics have yet to be developed. This paper proposes **Distribution Matching Policy Optimization (DMPO)**, a principled and theoretically grounded RL fine-tuning method specifically designed to enhance the reasoning capabilities of dLLMs by matching the dLLM policy distribution to the optimal, reward-tilted one through cross-entropy optimization. We identify a key implementation challenge with small training batch sizes and propose several effective solutions based on a novel weight baseline subtraction technique. DMPO exhibits superior performance on multiple reasoning benchmarks without supervised fine-tuning, achieving up to a 39.63 percentage-point improvement in accuracy over prior non-DMPO RL baselines and 67.97 percentage points over the base model, underscoring the effectiveness of the distribution-matching framework. Our code is available at https://github.com/yuchen-zhu-zyc/DMPO.

## 1. Introduction

Autoregressive large language models (AR-LLMs) have demonstrated remarkable capabilities in addressing sophisticated reasoning tasks, such as solving challenging math questions and completing coding tasks (Jaech et al., 2024; Anthropic, 2025; Guo et al., 2025a; Novikov et al., 2025; Kimi Team et al., 2025). While these models form their amazing capabilities from pretraining on massive text corpora, the main powerhouse behind the success is scaling the post-training phase with reinforcement learning (RL) techniques, such as Proximal Policy Optimization (PPO, Schulman et al. (2017)) and Group Relative Policy Optimization (GRPO, Shao et al. (2024)), which enhance model abilities through exploration of reward functions and go beyond static datasets. While possessing extraordinary competence, AR-LLMs are known to be expensive for inference due to their sequential, fixed left-to-right generation order, which currently prohibits large-scale deployment.

To address such issues, diffusion large language models (dLLMs) have been investigated as an alternative to the AR models. Unlike AR-LLMs, dLLMs iteratively refine a sequence from masked inputs, enabling any-order generation, and have shown promising performance on text modeling tasks. dLLMs such as LLaDA (Nie et al., 2025b) and Dream (Ye et al., 2025), have demonstrated competitive performances on many tasks compared to similar-size AR baselines. Recently, commercial models such as Mercury (Inception Labs et al., 2025) and Gemini Diffusion (DeepMind) have demonstrated the ability to achieve significantly higher inference throughput without sacrificing generation quality, suggesting dLLMs as a promising direction for language modeling. However, one question that remains largely unanswered is how to transfer the success of RL on LLMs to dLLMs, thereby further scaling up their capability.

Designing RL algorithms for dLLMs faces two major challenges. Due to the bidirectional nature of dLLMs, estimating the log probability of generated sequences is more expensive than for AR models, making it less favorable to naively adapt LLM post-training algorithms like GRPO to dLLMs, which heavily rely on this estimate. The GRPO-style algorithms also do not leverage dLLM's unique characteristic of having a *forward* noising process, as they are backward-only algorithms when using generated rollouts. Moreover, existing RL frameworks for enhancing LLM reasoning capabilities overly focus on reward maximization (Guo et al., 2025a; Liu et al., 2025b; Zheng et al., 2025a). By targeting

---
[*]Equal contribution [†]Equal advising [1]Georgia Institute of Technology. Correspondence to: Yuchen Zhu <yzhu738@gatech.edu>, Yongxin Chen <yongchen@gatech.edu>.

*Proceedings of the 43$^{rd}$ International Conference on Machine Learning*, Seoul, South Korea. PMLR 306, 2026. Copyright 2026 by the author(s).

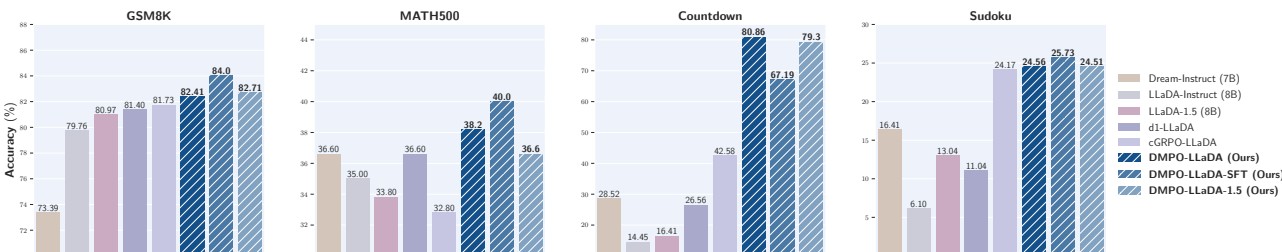

*Figure 1.* Performance on reasoning benchmarks evaluated with generation length 256. DMPO consistently achieves the best performance across dLLMs, outperforming prior RL baselines.

only the reward mode, these approaches do not fully leverage dLLMs' potential to generate diverse responses, given the task's random-order nature (Gong et al., 2026).

To jointly address these challenges, we propose **Distribution Matching Policy Optimization (DMPO)**, a principled and efficient RL fine-tuning method specifically designed for dLLMs. DMPO is designed on a novel framework grounded in stochastic optimal control (SOC), which shifts away from the conventional reward-maximization paradigm and targets the goal of matching the entire reward-tilted policy distribution. This enables the model to explore diverse, high-quality reasoning paths and responses during training, addressing concerns about over-focusing on absolute reward values and modes. In addition, DMPO training leverages importance sampling and a novel weighted denoising cross-entropy (WDCE) loss, which has the key advantage of operating in an **off-policy** manner, enabling the use of replay buffers to improve sample efficiency. More importantly, WDCE is a **forward-only** objective that relies solely on the obtained clean samples and the inexpensive, forward-noising process unique to dLLMs. DMPO largely decouples from rollout trajectories, potentially enabling it to achieve greater speed-ups than other dLLM RL algorithms when employed with fast inference techniques.

**Contributions** **(I)** We propose a novel RL learning framework for dLLMs that targets distribution matching rather than reward maximization (Sec. 3.1). **(II)** We propose Distribution Matching Policy Optimization (DMPO), a principled, theoretically-grounded fine-tuning strategy for enhancing dLLM's reasoning capabilities, supported by importance sampling and weighted denoising cross-entropy (Sec. 3.2). **(III)** We identify a special challenge that occurred for WDCE due to the use of a limited training batch size, and propose two novel techniques to address it: weight baseline subtraction (Sec. 3.3) and weighted direct discriminative optimization (Sec. 3.4). **(IV)** DMPO exhibits superior performances on multiple reasoning benchmarks without supervised fine-tuning (SFT), with an accuracy improvement up to 39.63 percentage points over prior non-DMPO RL baselines and 67.97 percentage points over the base model, being top-performing across bi-directional dLLMs (Sec. 4).

## 2. Preliminaries

### 2.1. Masked Diffusion Models for Language Modeling

The **masked (discrete) diffusion models (MDM)** (Lou et al., 2024; Ou et al., 2025; Sahoo et al., 2024; Shi et al., 2024; Zheng et al., 2025f) is a novel method for learning high-dimensional categorical distributions with application to text (Nie et al., 2025b), images (Chang et al., 2022; Bai et al., 2025), DNAs (Hayes et al., 2025), etc. Essentially, it learns the one-dimensional conditional distributions of the data given any subset of observed dimensions. Suppose the data are finite-length sequences with vocabulary $\mathcal{V} = \{1, 2, ..., V\}$. Include the mask token M into the $\mathcal{V}$ and let $\overline{\mathcal{V}} = \{1, 2, ..., V, \mathsf{M}\}$. The MDM takes a partially masked sequence $\boldsymbol{x} = (x_1, ..., x_D) \in \overline{\mathcal{V}}^D$ as an input, and outputs $\boldsymbol{\pi}_\theta(\boldsymbol{x}) \in \mathbb{R}^{D \times V}$, whose $(d, u)$-th entry $\boldsymbol{\pi}_\theta(\boldsymbol{x})_{d,u}$ is set to $1_{x_d=u}$ if $x_d \neq \mathsf{M}$, and if $x_d = \mathsf{M}$, is trained to approximate the conditional probability

$$\mathrm{Pr}_{\boldsymbol{X} \sim p_{\mathrm{data}}}(X_d = u | \boldsymbol{X}_{\mathrm{UM}} = \boldsymbol{x}_{\mathrm{UM}}),$$

where $\boldsymbol{x}_{\mathrm{UM}} = (x_d : x_d \neq \mathsf{M})$. By definition, we assume each row of $\boldsymbol{\pi}_\theta(\boldsymbol{x})$ is a valid probability vector. The probability of an unmasked sequence $\boldsymbol{x} \in \mathcal{V}^D$ under the MDM $\boldsymbol{\pi}_\theta$ is defined through **random-order autoregressive (ROAR) generation**: choosing a uniformly random order of the $D$ positions, and autoregressively sampling each position conditional on the previously sampled ones. Formally, let $\boldsymbol{\sigma}$ be a uniformly random permutation of $\{1, ..., D\}$, then

$$p_\theta(\boldsymbol{x}) = \mathbb{E}_{\boldsymbol{\sigma}} p_\theta(\boldsymbol{x}; \boldsymbol{\sigma}), \; p_\theta(\boldsymbol{x}; \boldsymbol{\sigma}) = \prod_{d=1}^{D} \boldsymbol{\pi}_\theta(x_{\sigma_d} | \boldsymbol{x}_{\boldsymbol{\sigma}_{<d}}). \quad (1)$$

Here, $\boldsymbol{\pi}_\theta(x_{\sigma_d} | \boldsymbol{x}_{\boldsymbol{\sigma}_{<d}})$ means input $\boldsymbol{x}$ with all positions except $\boldsymbol{\sigma}_{<d} = \{\sigma_1, ..., \sigma_{d-1}\}$ masked into the MDM and take the output at position $(\sigma_d, x_{\sigma_d})$.

The standard way to train an MDM given i.i.d. samples from $p_{\mathrm{data}}$ is to minimize the **denoising cross-entropy (DCE)** loss $\mathbb{E}_{p_{\mathrm{data}}(\boldsymbol{x})} \mathcal{L}_\theta(\boldsymbol{x})$, which involves the following definition of the (negative) **evidence lower bound (ELBO)** $\mathcal{L}_\theta$:

$$-\log p_\theta(\boldsymbol{x}) = -\log \mathbb{E}_{\boldsymbol{\sigma}} \, p_\theta(\boldsymbol{x}; \boldsymbol{\sigma}) \le -\mathbb{E}_{\boldsymbol{\sigma}} \log p_\theta(\boldsymbol{x}; \boldsymbol{\sigma})$$

$$= \underset{m \sim \mathrm{Unif}\{1,...,|\boldsymbol{x}|\}}{\mathbb{E}} \left[ \frac{|\boldsymbol{x}|}{m} \mathbb{E}_{\mu_m(\widetilde{\boldsymbol{x}}|\boldsymbol{x})} \sum_{d:\widetilde{x}_d = \mathsf{M}} -\log \boldsymbol{\pi}_\theta(\widetilde{\boldsymbol{x}})_{d,x_d} \right]$$

$$=: \mathcal{L}_\theta(\boldsymbol{x}), \tag{2}$$

where the transition distribution $\mu_m(\cdot|\boldsymbol{x})$ means to sample a uniformly random subset of $\{1,...,|\boldsymbol{x}|\}$ of size $m$ and mask the corresponding entries in $\boldsymbol{x}$, and $|\boldsymbol{x}|$ is the length of $\boldsymbol{x}$. The proof of the last equation can be found in Uria et al. (2016); Ou et al. (2025).

When applying to text data, the MDM is also referred to as the **diffusion large language model (dLLM)** (Nie et al., 2025b; Ye et al., 2025; Inception Labs et al., 2025; Song et al., 2025). For the purpose of reasoning, we typically write $\boldsymbol{x} = (\boldsymbol{q}, \boldsymbol{o})$, where $\boldsymbol{q}$ is the **prompt** (or query, which is always assumed to contain no mask state) and $\boldsymbol{o}$ is the **response** (or output). We use $\boldsymbol{\pi}_\theta(\boldsymbol{o}|\boldsymbol{q}) \in \mathbb{R}^{|\boldsymbol{o}| \times V}$ to denote the policy model output of the dLLM given a prompt $\boldsymbol{q}$ and a partially masked response $\boldsymbol{o}$. The conditional sequence probability of a clean model $\boldsymbol{o}$ given a prompt $\boldsymbol{q}$, denoted as $p_\theta(\boldsymbol{o}|\boldsymbol{q})$, is similarly defined through (1), where we now use notations $p_\theta(\boldsymbol{o}|\boldsymbol{q}; \boldsymbol{\sigma})$ and $\boldsymbol{\pi}_\theta(o_{\sigma_d}|\boldsymbol{q}, \boldsymbol{o}_{\sigma_{<d}})$ to emphasize the dependence on the prompt $\boldsymbol{q}$. The negative ELBO will be written as $\mathcal{L}_\theta(\boldsymbol{o}|\boldsymbol{q})$.

### 2.2. Reinforcement Learning for Enhancing Reasoning

We first present the **Group Relative Policy Optimization** (**GRPO**, Shao et al. (2024)) method for LLMs, which is the basis of most of the existing RL methods for dLLMs. Given a pretrained LLM with policy $\pi_{\mathrm{ref}}$ that samples from the distribution $p_{\mathrm{ref}}(\boldsymbol{o}|\boldsymbol{q}) = \prod_{d=1}^{|\boldsymbol{o}|} \pi_{\mathrm{ref}}(o_d|\boldsymbol{q}, \boldsymbol{o}_{<d})$, a reward function $r : (\boldsymbol{q}, \boldsymbol{o}) \mapsto \mathbb{R}$, a set of prompts $\mathcal{D}$, and a regularization parameter $\alpha \ge 0$, each step of the GRPO aims to solve the following problem: sample prompt $\boldsymbol{q} \sim \mathcal{D}$ and rollouts $\boldsymbol{o}^{(1:G)} \overset{\text{i.i.d.}}{\sim} p_{\theta_{\mathrm{old}}}(\boldsymbol{o}|\boldsymbol{q})$, then, maximize

$$\mathbb{E} \left\{ \frac{1}{G} \sum_{i=1}^{G} \frac{1}{|\boldsymbol{o}^{(i)}|} \sum_{d=1}^{|\boldsymbol{o}^{(i)}|} \left[ \min \left( \rho_d^{(i)} A_i, \mathrm{clip}(\rho_d^{(i)})_{1 \pm \epsilon} A_i \right) \right. \right.$$

$$\left. \left. - \alpha \, \mathrm{KL}(p_\theta(\boldsymbol{o}^{(i)}|\boldsymbol{q}) \| p_{\mathrm{ref}}(\boldsymbol{o}^{(i)}|\boldsymbol{q})) \right] \right\}, \tag{3}$$

where the advantages[1] are $A_i = r(\boldsymbol{q}, \boldsymbol{o}^{(i)}) - \mathrm{mean}(r(\boldsymbol{q}, \boldsymbol{o}^{(1:G)}))$, the per-token probability ratios are $\rho_d^{(i)} = \frac{\pi_\theta(o_d^{(i)}|\boldsymbol{q}, \boldsymbol{o}_{<d}^{(i)})}{\pi_{\theta_{\mathrm{old}}}(o_d^{(i)}|\boldsymbol{q}, \boldsymbol{o}_{<d}^{(i)})}$, and the KL regularization term is estimated similarly by the per-token probability ratios between $\pi_\theta$ and $\pi_{\mathrm{ref}}$. The clipping threshold $\epsilon$ prevents overly large policy updates.

---

[1]As suggested by Liu et al. (2025b), we list here the version without normalization by standard deviation.

While (3) works well for LLMs, it is not directly applicable to dLLMs due to mismatch between the *dLLM policy (model output)* $\boldsymbol{\pi}_\theta(\boldsymbol{o}|\boldsymbol{q})$ and the *sequence likelihood* $p_\theta(\boldsymbol{o}|\boldsymbol{q})$: unlike in LLMs where these two quantities are easily connected through the chain rule, it is generally non-trivial to compute the per-token probability given the dLLM model output, and only ELBO (2) is available as a surrogate. To tackle this issue, diffu-GRPO (Zhao et al., 2025a) proposed to fully mask all response positions and partially masks the prompt $\boldsymbol{q}$, and feed this sequence into the model to obtain the approximate probability $p_\theta(o_d|\boldsymbol{q})$. Next, the sequence probability $p_\theta(\boldsymbol{o}|\boldsymbol{q})$ is approximated by mean-field decomposition: $p_\theta(\boldsymbol{o}|\boldsymbol{q}) \approx \prod_{d=1}^{|\boldsymbol{o}|} p_\theta(o_d|\boldsymbol{q})$. Such approximations do not capture correlations among different positions in the response, leading to imprecision. A similar technique is employed in coupled-GRPO (cGRPO) for code generation tasks in Gong et al. (2026).

## 3. Distribution Matching Policy Optimization

### 3.1. Reward Maximization → Distribution Matching

To incentivize the reasoning capabilities of large language models, reward-maximizing reinforcement learning finetuning algorithms, such as TRPO (Schulman et al., 2015), PPO (Schulman et al., 2017), and GRPO (Shao et al., 2024), are often employed, with an additional entropy regularization term

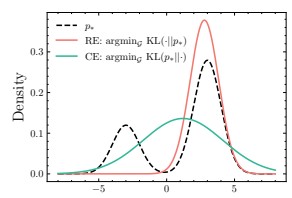

*Figure 2.* Illustration of relative entropy (mode-seeking) and cross-entropy (mass-covering) for fitting a target $p_*$ ($\mathcal{G}$ is the set of Gaussian distributions).

that penalizes the deviation of the model from the pretrained one. This process amounts to solving the following optimization problem,

$$\max_\theta \; \underset{\boldsymbol{q} \sim \mathcal{D}}{\mathbb{E}} \Big[ \underset{p_\theta(\boldsymbol{o}|\boldsymbol{q})}{\mathbb{E}} r(\boldsymbol{q}, \boldsymbol{o}) - \alpha \, \mathrm{KL}(p_\theta(\cdot|\boldsymbol{q}) \| p_{\mathrm{ref}}(\cdot|\boldsymbol{q})) \Big]. \tag{4}$$

However, existing techniques over-focus on finding and optimizing the **reward mode** and adopt many heuristic techniques to accelerate the mode searching process, neglecting the exploration of the entire distribution landscape, and often result in model mode collapse or reward hacking, causing the model to produce undesirable responses (Weng, 2024). A simple fix to this issue and to encourage diverse model responses is to enforce the optimality of the target policy distribution during the training. It can be shown that the optimal sequence distribution that solves the problem (4) is the following **reward-tilted distribution**:

$$p_*(\boldsymbol{o}|\boldsymbol{q}) = \frac{1}{Z(\boldsymbol{q})} p_{\mathrm{ref}}(\boldsymbol{o}|\boldsymbol{q}) \mathrm{e}^{r(\boldsymbol{q}, \boldsymbol{o})/\alpha}, \tag{5}$$

where $Z(\boldsymbol{q}) = \sum_{\boldsymbol{o}} p_{\mathrm{ref}}(\boldsymbol{o}|\boldsymbol{q}) \mathrm{e}^{r(\boldsymbol{q}, \boldsymbol{o})/\alpha}$. That is to say, we

want to use the optimal sequential distribution $p_*(\boldsymbol{o}|\boldsymbol{q})$ as the **supervision signal** throughout the learning process, so that we can learn a dLLM policy $\boldsymbol{\pi}_\theta$ which produces a sequence distribution $p_\theta$ matching $p_*$. We can thus obtain a policy that not only explores the dominant reward mode but is also guaranteed to sample other high-reward trajectories with a likelihood proportional to their reward values. This motivates us to consider the following task:

> **Policy Distribution Matching Learning:** Given a pre-trained dLLM policy $\boldsymbol{\pi}_{\mathrm{ref}}(\boldsymbol{o}|\boldsymbol{q})$ that samples from a distribution $p_{\mathrm{ref}}(\boldsymbol{o}|\boldsymbol{q})$, a reward function $r : (\boldsymbol{q}, \boldsymbol{o}) \mapsto \mathbb{R}$, a set of prompts $\mathcal{D}$, and temperature $\alpha > 0$, learn a dLLM policy $\boldsymbol{\pi}_\theta(\boldsymbol{o}|\boldsymbol{q})$ to produce the desired optimal distribution $p_*(\boldsymbol{o}|\boldsymbol{q})$ (5) by optimizing the following objective:
>
> $$\min_{\boldsymbol{\pi}_\theta} \mathbb{E}_{\boldsymbol{q} \sim \mathcal{D}} \, \mathcal{F}(p_\theta(\cdot|\boldsymbol{q}), p_*(\cdot|\boldsymbol{q})). \qquad (6)$$

Here, $\mathcal{F}$ is a class of functionals such that $\operatorname{argmin}_p \mathcal{F}(p, p_*) = p_*$. Note that the original entropy-regularized entropy optimization problem is equivalent to choosing $\mathcal{F}$ to be the reverse KL between $p$ and $p_*$, i.e., $\mathcal{F}(p_\theta, p_*) = \mathrm{KL}(p_\theta \| p_*) = \mathbb{E}_{p_\theta}[\log \frac{p_\theta}{p_*}]$. While this objective, in theory, can also yield the same optimal distribution with the desired property, reverse KL is widely known to be *mode-seeking*, i.e., it tends to match the dominant mode in $p_*$ while potentially neglecting other modes, which may lead to reward hacking.

To address this issue, we consider a series of new objectives $\mathcal{F}$ with more desirable convergence guarantees that steadily lead to optimization towards the desired sequence distribution, and propose **Distribution Matching Policy Optimization (DMPO)** (Alg. 1), which targets matching the entire reward-tilted policy distribution. In Sec. 3.2, we introduce **weighted denoising cross-entropy (WDCE)**, a **scalable** implementation of the forward KL using importance sampling. In Secs. 3.3 and 3.4, we discuss an important failure case of forward KL with **small training batch size**, and propose a series of novel techniques such as **weight baseline subtraction** (Sec. 3.3) and **weighted direct discriminative optimization** (Sec. 3.4) to address it.

### 3.2. Weighted Denoising Cross-entropy

Unlike the reverse KL objective considered by many existing works, which are known to be prone to mode seeking and collapse, one alternative choice is to use the forward KL divergence (or **cross-entropy, CE**) for the functional, i.e., $\mathcal{F}(p_\theta, p_*) = \mathrm{KL}(p_* \| p_\theta)$, which tends to cover all the modes of the optimal distribution and can retain the response diversity. The CE loss is widely used in another domain, stochastic optimal control (SOC) (Domingo-Enrich et al., 2024; 2025), which is closely connected to our work. This

amounts to solving the following task,

$$\min_\theta \mathbb{E}_{\boldsymbol{q} \sim \mathcal{D}} \, \mathbb{E}_{p_*(\boldsymbol{o}|\boldsymbol{q})} \left[ \log \frac{p_*(\boldsymbol{o}|\boldsymbol{q})}{p_\theta(\boldsymbol{o}|\boldsymbol{q})} \right]. \qquad (7)$$

However, objective (7) is not directly amenable to practical implementation, as we do not have access to real samples from the $p_*$, nor can we exactly compute $\log p_*$ due to the presence of the unknown partition function $Z(\boldsymbol{q})$. To bypass this issue, we draw inspiration from the recent work masked diffusion neural sampler (MDNS, Zhu et al. (2025e)), which proposed a training framework for learning a neural sampler based on MDM with stochastic optimal control and cross-entropy minimization. While targeting a different task, the core of MDNS resides in solving the same distribution matching problem with cross-entropy loss, and it proposed a practically implementable and scalable variant of (7) named **weighted denoising cross-entropy (WDCE)** loss. The central idea is to introduce a reference policy and leverage *importance sampling* to treat i.i.d. samples as importance-weighted samples from $p_*$. Taking advantage of this approach, we now derive WDCE for dLLM policy learning.

First, given the relationship between the policy output and sequence distribution of the masked dLLM (1), it is clear that we can match the correct target sequence distribution $p_*(\boldsymbol{o}|\boldsymbol{q})$ as long as we train $p_\theta(\boldsymbol{o}|\boldsymbol{q}; \boldsymbol{\sigma})$ to match the *order-specific* ones, i.e., $p_*(\boldsymbol{o}|\boldsymbol{q}; \boldsymbol{\sigma})$, given by

$$p_*(\boldsymbol{o}|\boldsymbol{q}; \boldsymbol{\sigma}) = \frac{1}{Z(\boldsymbol{q})} p_{\mathrm{ref}}(\boldsymbol{o}|\boldsymbol{q}; \boldsymbol{\sigma}) \mathrm{e}^{r(\boldsymbol{q}, \boldsymbol{o})/\alpha}. \qquad (8)$$

Leveraging this fact, given any prompt $\boldsymbol{q}$, we can express the cross-entropy loss as follows:

$$\begin{aligned}
\mathrm{KL}(p_*(\cdot|\boldsymbol{q}) \| p_\theta(\cdot|\boldsymbol{q})) &= \mathbb{E}_{p_*(\boldsymbol{o}|\boldsymbol{q})}[-\log p_\theta(\boldsymbol{o}|\boldsymbol{q})] + \mathrm{const} \\
&= \mathbb{E}_{\boldsymbol{\sigma}} \, \mathbb{E}_{p_*(\boldsymbol{o}|\boldsymbol{q}; \boldsymbol{\sigma})}[-\log p_\theta(\boldsymbol{o}|\boldsymbol{q})] + \mathrm{const} \\
&= \mathbb{E}_{\boldsymbol{\sigma}} \, \mathbb{E}_{p_v(\boldsymbol{o}|\boldsymbol{q}; \boldsymbol{\sigma})} \frac{p_*(\boldsymbol{o}|\boldsymbol{q}; \boldsymbol{\sigma})}{p_v(\boldsymbol{o}|\boldsymbol{q}; \boldsymbol{\sigma})}[-\log p_\theta(\boldsymbol{o}|\boldsymbol{q})] + \mathrm{const}, \qquad (9)
\end{aligned}$$

where $p_v$ is the sequence probability under a reference policy model $v$ that does not involve gradient computation, and in practice, one often chooses $v \leftarrow \bar{\theta} := \operatorname{stopgrad}(\theta)$ to be a copy of the policy model detached from the computation graph, and periodically synchronizes with the current model policy $p_\theta$, which is also commonly referred to as $p_{\theta_{\mathrm{old}}}$ in the literature. The importance weight $w(\boldsymbol{o}|\boldsymbol{q}; \boldsymbol{\sigma}) := \frac{p_*(\boldsymbol{o}|\boldsymbol{q}; \boldsymbol{\sigma})}{p_v(\boldsymbol{o}|\boldsymbol{q}; \boldsymbol{\sigma})}$ captures the mismatch between $p_v$ and $p_*$ and ensures the mathematical correctness of the objective, and $\log p_\theta(\boldsymbol{o}|\boldsymbol{q})$ is an intractable sequence log probability under the current dLLM policy. We discuss the computation of these two components in parallel below.

**Importance weight** $w(\boldsymbol{o}|\boldsymbol{q}; \boldsymbol{\sigma})$ We simplify it with the pretrained model and the reward:

$$w(\boldsymbol{o}|\boldsymbol{q};\boldsymbol{\sigma}) = \frac{1}{Z(\boldsymbol{q})} \frac{p_{\mathrm{ref}}(\boldsymbol{o}|\boldsymbol{q};\boldsymbol{\sigma})}{p_v(\boldsymbol{o}|\boldsymbol{q};\boldsymbol{\sigma})} \mathrm{e}^{\frac{r(\boldsymbol{q},\boldsymbol{o})}{\alpha}}$$

$$\propto \exp\left(\frac{r(\boldsymbol{q},\boldsymbol{o})}{\alpha} + \log \frac{p_{\mathrm{ref}}(\boldsymbol{o}|\boldsymbol{q};\boldsymbol{\sigma})}{p_v(\boldsymbol{o}|\boldsymbol{q};\boldsymbol{\sigma})}\right) =: \mathrm{e}^{\ell(\boldsymbol{o}|\boldsymbol{q};\boldsymbol{\sigma})}. \quad (10)$$

Recall that the order-specific probability of a sequence is computed via (1). To ensure that the sample distribution after importance sampling is valid and normalized, we keep track of the **log weights** $\ell(\boldsymbol{o}|\boldsymbol{q};\boldsymbol{\sigma})$, and take softmax among those corresponding to the same prompt $\boldsymbol{q}$ to compute the real weight $w(\boldsymbol{o}|\boldsymbol{q};\boldsymbol{\sigma})$. This is equivalent to estimating the unknown partition function $Z(\boldsymbol{q})$ using an empirical estimator of the following expectation:

$$Z(\boldsymbol{q}) = \mathbb{E}_{\boldsymbol{\sigma}} \mathbb{E}_{p_v(\boldsymbol{o}|\boldsymbol{q};\boldsymbol{\sigma})} \left[\frac{p_{\mathrm{ref}}(\boldsymbol{o}|\boldsymbol{q};\boldsymbol{\sigma})}{p_v(\boldsymbol{o}|\boldsymbol{q};\boldsymbol{\sigma})} \mathrm{e}^{r(\boldsymbol{q},\boldsymbol{o})/\alpha}\right].$$

The need to estimate partition functions is common in RL algorithms for LLM, such as in GflowNet (Bengio et al., 2021; Kimi Team et al., 2025). In contrast to these approaches that learn such functions independently, our estimation approach is training-free and more efficient.

**Sequence log probability** $\log p_v(\boldsymbol{o}|\boldsymbol{q})$   Unlike the case of LLM, the exact sequence log probability is intractable in dLLM due to the presence of expectation over the random order $\boldsymbol{\sigma}$. However, similar to the training of dLLM, we can leverage the negative ELBO (2) as a surrogate. Combined with the importance weight $w(\boldsymbol{o}|\boldsymbol{q};\boldsymbol{\sigma})$, we introduce the **weighted denoising cross-entropy (WDCE)** loss for dLLM policy distribution matching:

$$\min_{\theta} \mathbb{E}_{\boldsymbol{q}\sim\mathcal{D}} \mathbb{E}_{\boldsymbol{\sigma}} \mathbb{E}_{p_v(\boldsymbol{o}|\boldsymbol{q};\boldsymbol{\sigma})} \Big\{w(\boldsymbol{o}|\boldsymbol{q};\boldsymbol{\sigma}) \mathbb{E}_{m\sim\mathrm{Unif}\{1,\ldots,|\boldsymbol{o}|\}}\Big[$$
$$\frac{|\boldsymbol{o}|}{m} \mathbb{E}_{\mu_m(\widetilde{\boldsymbol{o}}|\boldsymbol{o})} \sum_{d:\widetilde{o}_d=\mathsf{M}} -\log \boldsymbol{\pi}_\theta(\widetilde{\boldsymbol{o}}|\boldsymbol{q})_{d,o_d}\Big]\Big\}. \quad (11)$$

Notably, this loss highly resembles the DCE loss used in pre-training and the supervised fine-tuning (SFT) phase of dLLM. One major difference is that instead of using i.i.d. samples from $p_*$, we use *importance sampling* to weight samples from $p_v$ and obtain a valid training objective with theoretical guarantees. WDCE differs significantly from other popular RL training techniques, such as PPO/GRPO, in the following two key aspects:

**WDCE is an *off-policy* loss**   The WDCE loss remains valid as the model parameter $\theta$ gets updated, since both the sampling policy $p_v$ and the important sampling target policy $p_*$ are independent of the current model policy $p_\theta$. This allows us to save generated rollouts in a replay buffer and reuse them across multiple training updates, without worrying excessively about numerical instability, thereby improving sample efficiency. On the other hand, for on-policy methods, to use a replay buffer, one would need

to estimate importance weights with respect to the current model policy $p_\theta(\boldsymbol{o}|\boldsymbol{q})$, i.e., $\frac{p_\theta(\boldsymbol{o}|\boldsymbol{q})}{p_v(\boldsymbol{o}|\boldsymbol{q})}$. Different from the case of LLM where such estimation can be done in one model forward pass, an accurate estimation in dLLM **per training update** is expensive, rendering the on-policy method less efficient. Moreover, for large models, where rollout generation and sequence likelihood estimation are typically handled by different implementations (such as vLLM and FSDP), this could lead to more nuanced, hard-to-detect biases that secretly undermine the algorithm's performance (Yao et al., 2025). With WDCE, we are largely free of such concerns.

**WDCE is a *forward* loss**   Unlike GRPO-style algorithms that typically require tracking the entire rollout trajectories, WDCE leverages the forward (noising) process during training, which is a characteristic *unique* to **diffusion** LLMs. Once we obtain the final samples and their associated weights, we can discard the trajectories and perform training using the cheap forward process by randomly masking the data. This implies that the training speed with WDCE largely depends on the model's inference speed. With the advances of dLLM efficient inference techniques such as fast decoding algorithms and KV-cache techniques (Ma et al., 2025; Hu et al., 2025; Wu et al., 2026; Liu et al., 2025c), WDCE could also enjoy a great boost in efficiency. This method also effectively leverages dLLM's potential to surpass LLMs in inference throughput, distinguishing it from other RL baselines that merely adapt LLM algorithms to dLLM. We defer a more detailed discussion of such properties to App. B.2.

**Stochastic optimal control (SOC) perspective**   While we developed WDCE through the lens of distribution matching (6), it can also be derived from a path-measure SOC viewpoint, following Wang et al. (2025a); Zhu et al. (2025e); we sketch the argument here and defer details to App. B.1. The random-order autoregressive sampling of $\boldsymbol{\pi}_\theta$ can be viewed as a continuous-time Markov chain (CTMC) on $\overline{\mathcal{V}}^D$ with rate matrix $Q_t^\theta(\boldsymbol{x}, \boldsymbol{x}^{d\leftarrow n}) = \gamma(t)\boldsymbol{\pi}_\theta(\boldsymbol{x})_{d,n}\mathbf{1}_{x^d=\mathsf{M}}$, inducing a **path measure** $\mathbb{P}^\theta$ whose terminal distribution coincides with $p_\theta$. To produce $p_*$ at the terminal time, we define the **target path measure**

$$\mathbb{P}^*(\boldsymbol{\xi}) := \tfrac{1}{Z}\mathbb{P}^{\mathrm{ref}}(\boldsymbol{\xi})\,\mathrm{e}^{r(\boldsymbol{\xi}_1)/\alpha},$$

which preserves the pretrained transition dynamics while tilting its terminal distribution to exactly $p_*$ in (5). A standard CTMC Radon–Nikodým calculation (see Lem. B.1) shows that diagonal and integral contributions cancel, leaving the closed-form path-level log-weight $\log \frac{\mathrm{d}\mathbb{P}^*}{\mathrm{d}\mathbb{P}^\theta}(\boldsymbol{\xi}) = W^\theta(\boldsymbol{\xi}) - \log Z$, where $W^\theta(\boldsymbol{\xi})$ is the trajectory log-weight defined in (22) and depends only on the terminal reward $r(\boldsymbol{\xi}_1)$ and on along-trajectory policy ratios $\frac{\boldsymbol{\pi}_{\mathrm{ref}}}{\boldsymbol{\pi}_\theta}$ at the jump

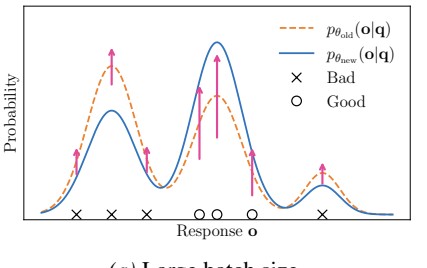

*(a)* Large batch size.

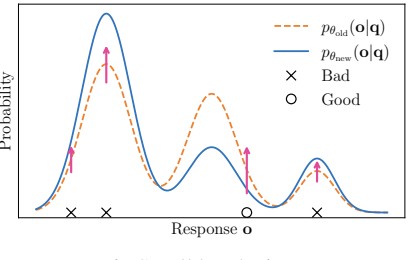

*(b)* Small batch size.

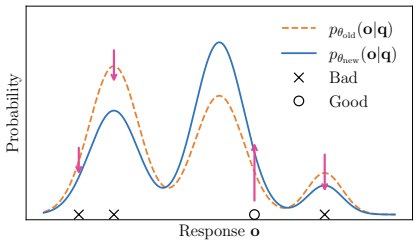

*(c)* Small batch size with baseline.

*Figure 3.* Demonstration of the effect of weight baseline. The orange and blue curves represent the probability $p_\theta(\boldsymbol{o}|\boldsymbol{q})$ before and after update, and the magenta arrows represent the weights. *(a)* When the batch size is large, the distribution mode coverage is good. Though bad responses have positive weights, the correct ones have larger weights to push the distribution updates in the right direction. *(b)* When the batch size is small, some modes (e.g., the good one in the middle) may not be sampled. Without **weight baseline subtraction**, the dominant positive weights of the bad responses lead to wrong update directions. *(c)* With **weight baseline subtraction**, the bad responses will appropriately be penalized, leading to the desired update direction.

times. Minimizing $\mathrm{KL}(\mathbb{P}^*\|\mathbb{P}^\theta)$ by importance sampling from a stop-gradient copy $\mathbb{P}^v$ gives

$$\mathrm{KL}(\mathbb{P}^*\|\mathbb{P}^\theta) = \mathop{\mathbb{E}}_{\mathbb{P}^v(\boldsymbol{\xi})} \frac{1}{Z} \mathrm{e}^{W^v(\boldsymbol{\xi})} \mathcal{L}_\theta(\boldsymbol{\xi}_1) + \mathrm{const},$$

which coincides *exactly* with the WDCE loss (11). Crucially, $\mathcal{L}_\theta$ is the *exact* negative log-density of $\mathbb{P}^\theta$ on the path space, so the ELBO *approximation* used at the sequence level in (9) corresponds to *precise* KL matching at the path level, justifying the validity of such surrogate.

### 3.3. Effective Training with Negative Gradient Insertion

While theoretically, minimizing the WDCE loss (11) provably leads to convergence of the model sequence distribution to $p_*(\boldsymbol{o}|\boldsymbol{q})$, this could face practical issues due to the often limited number of rollouts generated per prompt. Ideally, we would want to increase the likelihood of "good" responses while decreasing the likelihood of "bad" responses. However, with WDCE, any response $\boldsymbol{o}$ will be associated with a *positive* weight $w(\boldsymbol{o}|\boldsymbol{q};\boldsymbol{\sigma})$ due to the softmax operation, which may lead to ineffective learning in the low-batch-size scenario.

We note that this issue does not arise when the batch size is sufficiently large for the following reason. When having a large batch of diverse responses that make up a good coverage of the sample space, despite having all positive weights, since the model cannot increase likelihood on all responses (as the probabilities sum up to 1), the "bad" responses will be automatically and implicitly penalized due to not having larger weights than the "good" responses.

When the batch size is small, the scenario is different as is illustrated in Fig. 3. In such a case, the model will tend to **promote both "good" and "bad" responses** due to the positive weights, and potentially penalize the likelihood of other unseen responses to maintain a valid probability distribution. This could be detrimental to achieving distribution

matching, as these unseen responses may have high reward values and correspond to an undiscovered distribution mode.

To address this issue, we inject negative gradient (Ren & Sutherland, 2025; Deng et al., 2025) by designing a **weight baseline** and subtract it from the obtained weights to facilitate an effective reinforcement on the good samples, i.e.,

$$w_{\mathrm{real}}(\boldsymbol{o}|\boldsymbol{q};\boldsymbol{\sigma}) = w(\boldsymbol{o}|\boldsymbol{q};\boldsymbol{\sigma}) - w_{\mathrm{base}}(\boldsymbol{o}|\boldsymbol{q};\boldsymbol{\sigma}). \quad (12)$$

This approach resembles that adopted by PPO/GRPO. However, unlike these methods, we rate responses based on the log weights $\ell(\boldsymbol{o}|\boldsymbol{q};\boldsymbol{\sigma})$, where larger values indicate better alignment with the target optimal distribution. As a consequence, we promote responses that are more likely to be sampled from $p_*$ and penalize those that are less likely. Based on this perspective, we consider the following three methods for choosing $w_{\mathrm{base}}(\boldsymbol{o}|\boldsymbol{q};\boldsymbol{\sigma})$.

**Group weight baseline** When the dLLM policy is close to optimal, the original log weight $\ell(\boldsymbol{o}|\boldsymbol{q};\boldsymbol{\sigma})$ should behave approximately like constants for a group of different responses $\{\boldsymbol{o}^{(n)}\}_{1 \le n \le N}$, leading to nearly uniform weight value for $\{w(\boldsymbol{o}^{(n)}|\boldsymbol{q};\boldsymbol{\sigma}^{(n)})\}_{1 \le n \le N}$ after group softmax. We can thus choose the baseline as 1 to encourage convergence to this optimal situation:

$$w_{\mathrm{base}}(\boldsymbol{o}^{(n)}|\boldsymbol{q};\boldsymbol{\sigma}^{(n)}) = 1. \quad (13)$$

**Individual weight baseline** We can also consider the individual weight value of each response. For samples with smaller log weights, a stronger penalization is more desirable. A natural, adaptive way to design penalization strength is to use softmax over the log weights with *negative reward*: define $\ell_-(\boldsymbol{o}|\boldsymbol{q};\boldsymbol{\sigma}) := -\frac{r(\boldsymbol{q},\boldsymbol{o})}{\alpha} + \log \frac{p_{\mathrm{ref}}(\boldsymbol{o}|\boldsymbol{q};\boldsymbol{\sigma})}{p_v(\boldsymbol{o}|\boldsymbol{q};\boldsymbol{\sigma})}$, and

$$w_{\mathrm{base}}(\boldsymbol{o}^{(n)}|\boldsymbol{q};\boldsymbol{\sigma}^{(n)}) = \frac{N \exp(\ell_-(\boldsymbol{o}^{(n)}|\boldsymbol{q};\boldsymbol{\sigma}^{(n)}))}{\sum_k \exp(\ell_-(\boldsymbol{o}^{(k)}|\boldsymbol{q};\boldsymbol{\sigma}^{(k)}))}. \quad (14)$$

Note that this $w_{\mathrm{base}}(\boldsymbol{o}|\boldsymbol{q};\boldsymbol{\sigma})$ now corresponds to a *bad target distribution* given by $p_{*-}(\boldsymbol{o}|\boldsymbol{q}) \propto_{\boldsymbol{o}} p_{\mathrm{ref}}(\boldsymbol{o}|\boldsymbol{q})\mathrm{e}^{-r(\boldsymbol{q},\boldsymbol{o})/\alpha}$,

which is tilted by the negative reward. The minus sign in the loss before $w_{\text{base}}(\boldsymbol{o}|\boldsymbol{q};\boldsymbol{\sigma})$ means we want to drive the dLLM policy away from this bad distribution.

**Model weight baseline** Finally, we can determine whether to promote or penalize specific responses by comparing $w(\boldsymbol{o}|\boldsymbol{q};\boldsymbol{\sigma})$ with the importance weight under the current model policy $p_\theta(\boldsymbol{o}|\boldsymbol{q})$, which pushes the model further towards the optimal $p_*(\boldsymbol{o}|\boldsymbol{q})$. Note that this does not incur additional computation overhead as we can estimate $\log p_{\bar{\theta}}(\boldsymbol{o}|\boldsymbol{q})$ using negative ELBO (2), which is already computed in the WDCE loss. Define $\ell_\theta(\boldsymbol{o}|\boldsymbol{q};\boldsymbol{\sigma}) := \log\frac{p_{\bar{\theta}}(\boldsymbol{o}|\boldsymbol{q};\boldsymbol{\sigma})}{p_v(\boldsymbol{o}|\boldsymbol{q};\boldsymbol{\sigma})}$, and let

$$w_{\text{base}}(\boldsymbol{o}^{(n)}|\boldsymbol{q};\boldsymbol{\sigma}^{(n)}) = \frac{N\exp(\ell_\theta(\boldsymbol{o}^{(n)}|\boldsymbol{q};\boldsymbol{\sigma}^{(n)}))}{\sum_k \exp(\ell_\theta(\boldsymbol{o}^{(k)}|\boldsymbol{q};\boldsymbol{\sigma}^{(k)}))}. \quad (15)$$

We remark that the group weight and model weight baselines (13) and (15) can also be interpreted as an *approximate variance reduction*. See App. B.3 for discussion.

### 3.4. Weighted Direct Discriminative Optimization

To explore the full potential of the distribution matching framework in (6), we also investigate other choices for the potential $\mathcal{F}$ beyond the cross-entropy. One particularly interesting objective is the following **direct discriminative optimization (DDO)** loss:

$$\mathcal{F}(p_\theta(\cdot|\boldsymbol{q}), p_*(\cdot|\boldsymbol{q})) = -\mathbb{E}_{p_*(\boldsymbol{o}|\boldsymbol{q})}\log\sigma\Big(\log\frac{p_\theta(\boldsymbol{o}|\boldsymbol{q})}{p_v(\boldsymbol{o}|\boldsymbol{q})}\Big)$$
$$- \mathbb{E}_{p_v(\boldsymbol{o}|\boldsymbol{q})}\log\sigma\Big(-\log\frac{p_\theta(\boldsymbol{o}|\boldsymbol{q})}{p_v(\boldsymbol{o}|\boldsymbol{q})}\Big), \quad (16)$$

where $\sigma(t) = 1/(1+e^{-t})$. The global minimizer of (16) is also $p_*(\cdot|\boldsymbol{q})$, thus being a valid functional for distribution matching. For a more detailed justification, see App. B.4.

This is inspired by Zheng et al. (2025e), which proposed a GAN-like (Goodfellow et al., 2020) loss for SFT of vision models. One interesting trait of this objective is its natural incorporation of negative gradients for bad samples due to the GAN nature, as is shown in the analysis therein:

$$\nabla_\theta\mathcal{F}(p_\theta(\cdot|\boldsymbol{q}), p_*(\cdot|\boldsymbol{q})) = \sum_{\boldsymbol{o}}\sigma\Big(-\log\frac{p_\theta(\boldsymbol{o}|\boldsymbol{q})}{p_v(\boldsymbol{o}|\boldsymbol{q})}\Big)$$
$$\cdot \boxed{(p_\theta(\boldsymbol{o}|\boldsymbol{q}) - p_*(\boldsymbol{o}|\boldsymbol{q}))}\nabla_\theta\log p_\theta(\boldsymbol{o}|\boldsymbol{q}).$$

From the expression, as the first term is always non-negative, and the $\boxed{\text{boxed}}$ term applies a penalty for bad response $\boldsymbol{o}$, thus providing a gradient direction for increasing $p_\theta(\boldsymbol{o}|\boldsymbol{q})$. Leveraging this property, we adapt it for RL finetuning of dLLM and introduce **weighted direct discriminative optimization (WDDO)** loss, again dealing with $p_*$ through *importance sampling*:

$$\mathcal{F}(p_\theta(\cdot|\boldsymbol{q}), p_*(\cdot|\boldsymbol{q})) = -\mathbb{E}_{\boldsymbol{\sigma}}\ \mathbb{E}_{p_v(\boldsymbol{o}|\boldsymbol{q};\boldsymbol{\sigma})}\Big[w(\boldsymbol{o}|\boldsymbol{q};\boldsymbol{\sigma})$$
$$\log\sigma\Big(\log\frac{p_\theta(\boldsymbol{o}|\boldsymbol{q})}{p_v(\boldsymbol{o}|\boldsymbol{q})}\Big) + \log\sigma\Big(-\log\frac{p_\theta(\boldsymbol{o}|\boldsymbol{q})}{p_v(\boldsymbol{o}|\boldsymbol{q})}\Big)\Big],$$

where $w(\boldsymbol{o}|\boldsymbol{q};\boldsymbol{\sigma})$ is the importance weight defined in (10).

## 4. Experiments

**Model and baselines** We apply DMPO to LLaDA-8B-Instruct (Nie et al., 2025b), a SOTA fully-bidirectional dLLM not post-trained with RL techniques. To clearly demonstrate the potential of DMPO, we follow an R1-Zero-like training recipe (Guo et al., 2025a; Liu et al., 2025b) and apply DMPO directly to the LLaDA and Dream models without first performing SFT on curated datasets. We refer to the model obtained from this pipeline as **DMPO-LLaDA** and **DMPO-Dream**. We benchmark our method against a series of dLLM base models of comparable size, such as Dream-Instruct (7B, Ye et al. (2025)), LLaDA-Instruct (8B, Nie et al. (2025b)), and LLaDA-1.5 (8B, Zhu et al. (2025a)). We compare with d1 (Zhao et al., 2025a), a dLLM RL fine-tuning approach that combines both SFT and diffu-GRPO (an adapted version of GRPO), as well as cGRPO (Gong et al., 2026), wd1 (Tang et al., 2025c), and GDPO (Rojas et al., 2025a).

**Experimental setups** We explore four reasoning benchmarks: GSM8k, MATH500, Sudoku, and Countdown. For all pretrained dLLMs, we evaluate the latest available checkpoint. For d1 and cGRPO, we reproduce their results exactly following the provided guidelines. To ensure a fair comparison, we train DMPO-LLaDA and DMPO-Dream on the same datasets as d1 for each task with rollouts generated using a fixed sequence length of 256. Evaluations are conducted with zero-shot prompting using generation lengths 128, 256, and 512 for LLaDA-series models, and 256 and 512 for Dream.

**DMPO incentivizes superior reasoning capabilities** We report in Tabs. 1 and 2 the performance of DMPO together with that of base dLLMs and models obtained by existing RL post-training strategies. DMPO consistently outperforms both the base models and the models tuned with d1/cGRPO/wd1/GDPO. On LLaDA, it achieves excellent gains over LLaDA-Instruct, with an accuracy improvement of an average of $+\mathbf{2.40\%}$ on GSM8K, $+\mathbf{3.00\%}$ on MATH500, $+\mathbf{59.38\%}$ on Countdown, and $+\mathbf{42.27\%}$ on Sudoku. Compared with the strongest non-DMPO RL baseline at each setting, DMPO-LLaDA achieves especially large gains on planning tasks, including $+\mathbf{13.67\%}$ and $+\mathbf{16.40\%}$ on Countdown, and $+\mathbf{39.63\%}$ and $+\mathbf{31.57\%}$ on Sudoku at generation lengths 256 and 512, respectively. On Dream-Instruct (7B), DMPO-Dream also outperforms

*Table 1.* Model performances on reasoning benchmarks for LLaDA-Instruct (8B). **Best** and second best results are highlighted. DMPO consistently outperforms other baselines across different generation length.

| Task | GSM8K | | | MATH500 | | | Countdown | | | Sudoku | | |
|---|---|---|---|---|---|---|---|---|---|---|---|---|
| Sequence Length | 128 | 256 | 512 | 128 | 256 | 512 | 128 | 256 | 512 | 128 | 256 | 512 |
| Dream-Instruct (7B) | 56.63 | 73.39 | 76.65 | 31.00 | 36.60 | 36.40 | 22.66 | 28.52 | 27.34 | 14.45 | 16.41 | 11.77 |
| LLaDA-Instruct (8B) | 71.87 | 79.76 | 83.62 | 28.20 | 35.00 | 38.80 | 23.44 | 14.45 | 14.84 | 12.94 | 6.10 | 7.37 |
| LLaDA-1.5 (8B) | 73.09 | 80.97 | 84.38 | 26.80 | 33.80 | 40.00 | 26.17 | 16.41 | 23.83 | 15.19 | 13.04 | 8.98 |
| d1-LLaDA | 75.28 | 81.40 | 84.38 | 30.00 | 36.60 | 40.80 | 34.38 | 26.56 | 30.47 | 21.97 | 11.04 | 8.69 |
| cGRPO-LLaDA | 67.40 | 81.73 | 84.23 | 21.40 | 32.80 | 38.40 | 30.08 | 42.58 | 37.11 | 24.17 | 24.17 | 21.97 |
| wd1-LLaDA | – | 80.80 | 82.31 | – | 37.60 | 39.80 | – | 49.22 | 47.17 | – | 22.04 | 24.65 |
| GDPO-LLaDA | – | 81.20 | 82.26 | – | 38.00 | 38.20 | – | 67.19 | 66.41 | – | 24.17 | 25.10 |
| **DMPO-LLaDA (Ours)** | 74.83 | 82.41 | **85.22** | 30.00 | 38.20 | **42.80** | 67.19 | 80.86 | 82.81 | **32.76** | **63.80** | **56.67** |
| **DMPO-LLaDA-SFT (Ours)** | **80.06** | **84.00** | 84.09 | **31.80** | **40.00** | 41.20 | 54.69 | 67.19 | 77.34 | 25.20 | 25.73 | 23.78 |
| **DMPO-LLaDA-1.5 (Ours)** | 77.56 | 82.71 | 84.61 | 30.20 | 36.60 | 41.00 | 59.77 | 79.30 | **83.20** | 25.34 | 24.51 | 23.34 |

*Table 2.* Model performances on reasoning benchmarks for Dream-Instruct (7B). **Best** and second best results are highlighted. DMPO consistently outperforms other baselines across different generation length.

| Task | GSM8K | | MATH500 | | Countdown | | Sudoku | |
|---|---|---|---|---|---|---|---|---|
| Sequence Length | 256 | 512 | 256 | 512 | 256 | 512 | 256 | 512 |
| Dream-Instruct (7B) | 73.39 | 76.65 | 36.60 | 36.40 | 28.52 | 27.34 | 16.41 | 11.77 |
| d1-Dream[†] | 80.52 | 82.41 | 41.20 | 45.60 | 29.46 | 36.83 | 21.78 | 23.34 |
| **DMPO-Dream (Ours)** | **84.03** | **84.76** | **47.40** | **47.20** | **54.43** | **56.51** | **41.04** | **45.55** |

d1-Dream across all tasks, with average gains of $+\mathbf{2.93}\%$ on GSM8K, $+\mathbf{3.90}\%$ on MATH500, $+\mathbf{22.32}\%$ on Countdown, and $+\mathbf{20.73}\%$ on Sudoku. This underscores the overall effectiveness of DMPO for enhancing model reasoning capabilities across different dLLM backbones.

**Weight baseline subtraction is crucial for small batch size training** We test different choices for negative gradient insertion in Secs. 3.3 and 3.4 when training on the Sudoku dataset with a small batch size, and the results are visualized in Fig. 4. As shown by the curves, without weight baseline subtraction, the model does not improve as training progresses. All the proposed weight baselines in (13) to (15) effectively increase the reward value during training. WDDO achieves the fastest reward increase during the initial 1k steps but suffers from instability afterwards.

**DMPO benefits from other means of post-training techniques** To showcase the robustness and general efficacy of DMPO, we apply it to LLaDA-SFT and LLaDA-1.5. LLaDA-SFT is obtained by performing SFT of LLaDA-Instruct (8B) on s1k (Muennighoff et al., 2025), a dataset of 1k examples of high-quality reasoning questions with distilled reasoning traces from Gemini Thinking;. LLaDA-

1.5 is obtained by performing DPO on 350K preference pairs covering a wide range of topics such as writing and reasoning. We then apply DMPO to these base models to obtain DMPO-LLaDA-SFT and DMPO-LLaDA-1.5, with performance reported in Tab. 1. DMPO continues to deliver performance gains for post-trained models, with consistent and significant accuracy improvements over base models, especially at generation lengths of 128 and 256 for the math reasoning datasets, with $+\mathbf{4.78}\%$ and $+\mathbf{2.60}\%$ on GSM8K, $+\mathbf{1.80}\%$ and $+\mathbf{3.40}\%$ on MATH500 compared with d1. This underscores that DMPO is a powerful method that integrates smoothly with existing solutions.

**DMPO enables efficient and fast training** Due to its *off-policy* and *forward* nature, DMPO achieves considerable training acceleration compared with GRPO-type methods. In Fig. 5, we compare head-to-head the training dynamics of diffu-GRPO, DMPO with ROAR sampler, and DMPO with Fast-dLLM (an approximate KV-cache mechanism enabled, confidence-based heuristic sampler for dLLMs, from Wu et al. (2026)) on Countdown under the same amount of training compute. Due to its off-policy nature, DMPO enables heavy reuse of each sampled buffer of rollouts and achieves sample efficiency $\mathbf{2} \sim \mathbf{3}\times$ higher than that of diffu-GRPO.

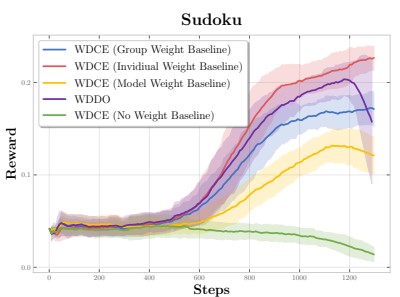

*Figure 4.* Effects of negative gradient insertion on Sudoku.

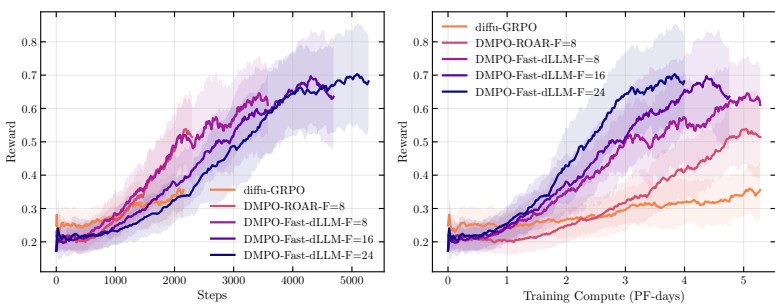

*Figure 5.* Comparison of training dynamics on Countdown. $F$ is the frequency of sampling the buffer.

Regarding training-compute efficiency, as a forward-loss-based algorithm, DMPO offers flexibility in the choice of rollout sampler. With fast-dLLM, DMPO gains an acceleration of up to $8\times$ per rollout sampling, and achieves the same level of reward as d1 with only $31\%$ of the training budget (1.8 PF-days [2] v.s. 5.78 PF-days). This empirical evidence emphasizes that DMPO is not only an effective algorithm but also highly sample- and compute-efficient.

**DMPO exhibits stable training despite highly stale data** As is evident from Figs. 5 and 6, DMPO enjoys a largely stable dynamics despite using up to $24\times$ stale data (which means 24 parameter updates on the same batch of rollouts), without suffering from high variance of importance sampling. While this seems to contradict the general belief that on-policy learning beats off-policy learning for LLM RL, we argue that this is *not* the case, because the off-policy in DMPO is inherently different from that used in diffu-GRPO or GRPO. Note that the latter considers the importance weight of the form $\frac{\pi_\theta}{\pi_{\text{old}}}$, which **inevitably diverges** as the number of parameter updates on $\theta$ increases. However, DMPO uses importance weight of the form $\frac{p_*}{p_{\text{old}}}$, which is independent of the current policy model $\pi_\theta$ and remains stable over a long horizon of training, enabling the use of a low buffer sampling frequency and highly stale rollouts without sacrificing performances. Moreover, DMPO adopts **sequence-level importance sampling**, in contrast to the *token-level* importance sampling used in diffu-GRPO or cGRPO, thereby providing an additional layer of stability. This advantage is also discussed in depth in Group Sequence Policy Optimization (GSPO, Zheng et al. (2025a)), which similarly considers *sequence-level* importance sampling.

Additional results and discussion can be found in App. C.3.

## 5. Conclusion

We propose Distribution Matching Policy Optimization (DMPO), a novel RL fine-tuning framework for dLLMs

that leverages their unique characteristics via importance sampling and a WDCE loss, enabling off-policy training and forward-only computation that naturally exploits dLLM inference capabilities. The main limitation of this work is that we focus on two pretrained dLLMs and four elementary reasoning benchmarks, and DMPO's performance on other pretrained dLLMs and tasks in different domains remains unknown. Our work opens several directions for future research, such as investigating the distribution-matching framework for other generative models and studying the design of more effective weight-baseline techniques.

## Acknowledgements

The authors are grateful for partial supports by NSF Grants ECCS-1942523, DMS-2206576, 2450378 (WG & YC), AFOSR Grant FA9550-25-1-0169 (WG & YC), Georgia Tech ARC-ACO Fellowship (WG), NSF Grant DMS-2513699 (YZ & MT), DOE Grants NA0004261 (MT), SC0026274 (YZ & MT), Richard Duke Fellowship (YZ & MT), and Simons Institute for the Theory of Computing at UC Berkeley (MT). This research was supported in part through research cyberinfrastructure resources and services provided by the Partnership for an Advanced Computing Environment (PACE) at the Georgia Institute of Technology, Atlanta, Georgia, USA. RRID:SCR_027619.

## Impact Statement

This paper presents work whose goal is to advance the field of Machine Learning. There are many potential societal consequences of our work, none which we feel must be specifically highlighted here.

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

---

**Algorithm 1** Distribution Matching Policy Optimization

---

**Require:** Training dataset $\mathcal{D}$, number of prompts per batch $B$, number of rollouts per prompt $N$, frequency for sampling buffer $F$, model policy $\pi_\theta$.
1: **for** step $= 0, 1, 2, ...$ **do**
2:    **if** step mod $F = 0$ **then**
3:       {Prepare the buffer using the current policy, denoted $\pi_v$.}
4:       Sample $B$ prompts $\{q^{(i)}\}_{1 \le i \le B}$ from the dataset $\mathcal{D}$.
5:       **for** $1 \le i \le B$ (in parallel, with gradient computation disabled) **do**
6:          Sample $N$ orders and generate $N$ rollouts $\{o^{(i,n)}\}_{1 \le n \le N}$ conditional on prompt $q^{(i)}$.
7:          Evaluate reward and compute weights $w(o^{(i,n)}|q^{(i)}; \sigma^{(i,n)})$ according to (10).
8:          Compute the weight baseline according to (13), (14), or (15), and obtain the real weights $w_{\text{real}}(o^{(i,n)}|q^{(i)}; \sigma^{(i,n)})$ according to (12).
9:       **end for**
10:    **end if**
11:    For each $o^{(i,n)}$, sample a mask assignment and obtain $\widetilde{o}^{(i,n)}$.
12:    Feed all pairs of $(q^{(i)}, \widetilde{o}^{(i,n)})$ into $\pi_\theta$ and compute the WDCE loss (11), then update $\theta$.
13: **end for**
**output** $\pi_\theta$

---

# A. Related Work

Here, we focus on the literature for discrete diffusion models, as well as the methods for fine-tuning MDMs, dLLMs, and LLMs. We also briefly review several GRPO-style algorithms for domains outside of LLMs.

**Discrete diffusion models**    Diffusion models have been top-performing approaches for generating various data modalities (Zhu et al., 2025d; Esser et al., 2024; Zhu et al., 2025b; Rojas et al., 2025b; Zheng et al., 2025e; Chen et al., 2025a; Ren et al., 2025c). Discrete diffusion models (Austin et al., 2021; Campbell et al., 2022; Lou et al., 2024; Zhang et al., 2025b), a natural extension of diffusion models to finite state spaces, have emerged as powerful approaches for generating categorical, sequence data, with applications to text (Nie et al., 2025a;b; Ye et al., 2025), images (Chang et al., 2022; Bai et al., 2025; Shi et al., 2025), and biological sequences (Tang et al., 2025a; Chen et al., 2025b). One of the most effective variants of discrete diffusion models is masked diffusion models (MDM) (Sahoo et al., 2024; Ou et al., 2025; Shi et al., 2024) and its variants (Arriola et al., 2025; Sahoo et al., 2025; Chao et al., 2025). Recently, continuous latents have also been introduced into the modeling of discrete data (Zhang et al., 2025a; Zhou et al., 2025; Zheng et al., 2025b), resulting in improved and more appealing performance.

One particularly important line of development for discrete diffusion models centers on their inference techniques, with the aim of improving generation quality (Nisonoff et al., 2025; Rojas et al., 2026; Kim et al., 2025; Besnier et al., 2025) and accelerating sampling speed (Ren et al., 2025b; Ben-Hamu et al., 2025; Wu et al., 2026; Hong et al., 2025). Besides these training-free approaches, learning-based approaches, such as few-step distillation, have also achieved decent success for discrete diffusion models (Deschenaux & Gulcehre, 2025; Karimi Monsefi et al., 2025; Zheng et al., 2025c; Zhu et al., 2025f;g). DMPO is closely tied to the literature on fast inference, as it can benefit from it by enjoying a similar training speed acceleration due to its forward nature.

**Fine-tuning general discrete diffusion models**    Earlier works on fine-tuning discrete diffusion models primarily focus on applications in biological and chemical domains, e.g., SVDD (Li et al., 2025), DDPP (Rector-Brooks et al., 2025), DRAKES (Wang et al., 2025a), SEPO (Zekri & Boullé, 2025), and TR2-D2 (Tang et al., 2025b). Although these methods work well for their respective tasks, they are not directly applicable to dLLMs due to the unique challenges posed by the language domain, such as large model size, high dimensionality, and the need to maintain linguistic coherence and diversity.

**Fine-tuning diffusion LLMs**    Recently, numerous works have proposed RL algorithms for fine-tuning dLLMs, with most existing works being adaptations of the GRPO algorithm (Shao et al., 2024) for AR LLMs. For example, Zhao et al. (2025a) proposed Diffu-GRPO that estimates the per-token response log probabilities via masking all except the required response positions, and partially masking the prompt to get the model output, while their sequence log probability is estimated

by mean-field approximation. Gong et al. (2026) introduced Coupled GRPO that modified the Diffu-GRPO method by not partially masking the prompt, and using complementary pairs of masks to mask the same response that fully uses the model output, which we also adopt in our experiments. Yang et al. (2025) proposed UniGRPO, which involves a structured noise strategy and a modified log-likelihood approximation (both per-token and sequence). Concurrent with our work, TraceRL (Wang et al., 2025b) improves dLLM RL training by minimizing a training-inference gap. wd1 (Tang et al., 2025c) introduces additional regularization to the old policy, alongside the regularization applied to the reference model policy, which resembles the case discussed in App. B.2. We highlight that all these methods are GRPO-style algorithms that require estimating per-token response log probabilities, which are typically intractable and challenging for dLLMs. In contrast, our method offers the advantage of being a forward one, with greater efficiency and accuracy.

**Fine-tuning LLMs** For fine-tuning LLMs, pre-LLM era works such as Trust Region Policy Optimization (TRPO, Schulman et al. (2015)) and Proximal Policy Optimization (PPO, Schulman et al. (2017)) have been widely used for RLHF (Ouyang et al., 2022). Since the huge success of GRPO (Shao et al., 2024) on DeepSeek-R1 (Guo et al., 2025a), there have been many follow-up works that improve GRPO in various ways, for instance: GRPO Done Right (Dr-GRPO, Liu et al. (2025b)), Decoupled clip and Dynamic sAmpling Policy Optimization (DAPO, Yu et al. (2025)), Group Policy Gradient (GPG, Chu et al. (2025)), Group Sequence Policy Optimization (GSPO, Zheng et al. (2025a)), Geometric-Mean Policy Optimization (GMPO, Zhao et al. (2025b)), etc.

Apart from the aforementioned policy gradient-based methods, GFlowNet (Bengio et al., 2021) has also been applied to finetuning LLMs, with successful applications seen in Kimi 1.5 (Kimi Team et al., 2025) and FlowRL (Zhu et al., 2025c). Notably, concurrent with our work, FlowRL shares the same high-level goal as our DMPO, targeting also policy distribution matching rather than merely reward maximization for AR-LLMs. However, distinct from DMPO, FlowRL derives its objectives from reverse KL and utilizes GFlowNet objectives. In contrast, our approach considers forward KL, which is known to be mass-covering, and implements it using importance sampling and weighted denoising cross-entropy.

**GRPO-style algorithms for fine-tuning diffusion and flow-based models** GRPO-type algorithms have also been adapted to diffusion and flow-based models, such as flow-GRPO (Liu et al., 2025a) and DanceGRPO (Xue et al., 2025). Aside from that, there are also SOC-based fine-tuning algorithms for diffusion models, such as adjoint matching (Domingo-Enrich et al., 2025), with which our work shares similarity at a high level. Concurrent with our work, DiffusionNFT (Zheng et al., 2025d) has been proposed to finetune continuous diffusion models for text-to-image generation tasks. While formulated in drastically different ways, DiffusionNFT shares a similarity with our DMPO in that it is also an algorithm that primarily depends on model forward passes rather than backward trajectories.

## B. Theory of Distribution Matching Policy Optimization

### B.1. Distribution Matching Policy Optimization from the Stochastic Optimal Control Perspective

This section aims at providing an alternative derivation of DMPO from the perspective of stochastic optimal control (SOC), which is inspired by DRAKES (Wang et al., 2025a) and MDNS (Zhu et al., 2025e). We will first introduce the necessary background on continuous-time Markov chains (CTMCs), then show how MDM sampling can be viewed as a CTMC. Finally, we derive the DMPO framework from the SOC perspective.

**Introduction to Continuous-time Markov Chains** To derive the SOC framework for fine-tuning, we view the sampling of an MDM as a time-indexed stochastic process, and the proper mathematical tool is the **continuous-time Markov chain (CTMC)**. A CTMC $X = (X_t)_{t \in [0,1]}$ is a stochastic process taking value in a discrete state space $\mathcal{X}$. Its law is characterized by the **rate matrix** $Q = (Q_t)_{t \in [0,1]}$, defined as

$$Q_t(x, y) = \lim_{\Delta t \to 0} \frac{\Pr(X_{t+\Delta t} = y | X_t = x) - 1_{x=y}}{\Delta t}, \ \forall x, y \in \mathcal{X}. \tag{17}$$

By definition, the off-diagonal entries of $Q_t$ are non-negative, and each row sums to zero.

The **path** of $X$, i.e., $t \mapsto X_t(\omega)$, is piecewise constant with discontinuous jumps, and one typically assumes that the path is right continuous with left limits. The **path measure** a CTMC $X$ is a probability measure on the space of paths defined as $\mathbb{P}^X(A) := \Pr(X \in A)$, which is the distribution of $X$. The following lemma shows how to compute the **Radon-Nikodým (RN) derivative** between two path measures driven by CTMCs with different rate matrices and initial distributions:

**Lemma B.1.** *Given two CTMCs with rate matrices $Q^1, Q^2$ and initial distributions $\mu_1, \mu_2$ on $\mathcal{X}$, let $\mathbb{P}^1, \mathbb{P}^2$ be the associated path measures. Then, for any path $\xi = (\xi_t)_{t \in [0,1]}$,*

$$\log \frac{d\mathbb{P}^1}{d\mathbb{P}^2}(\xi) = \log \frac{d\mu_1}{d\mu_2}(\xi_0) + \sum_{t:\xi_{t-} \neq \xi_t} \log \frac{Q_t^1(\xi_{t-}, \xi_t)}{Q_t^2(\xi_{t-}, \xi_t)} + \int_0^1 (Q_t^1(\xi_t, \xi_t) - Q_t^2(\xi_t, \xi_t))dt. \tag{18}$$

For the proof, see Campbell et al. (2024, App. C.1), Ren et al. (2025a, Thm. 3.3), or Zhu et al. (2025e, Lem. 1). An intuitive interpretation of (18) is to view the RN derivative as the limit of density ratios between finite-dimensional joint distributions, and approximate the transition probability by (17).

**Masked Diffusion Models as Continuous-Time Markov Chains** We will now delve into the CTMC formulation of sampling from an MDM. To avoid notational clutter, we use superscript to denote the position index, and subscript to denote the time index (e.g., $\boldsymbol{\xi}_t = (\xi_t^1, ..., \xi_t^D)$). We present the theory only in the case of *unconditional generation* with sequence length $D$ for simplicity of notations, but it can be easily easily generalized to the case of conditional generation of $\boldsymbol{o}$ given a prompt $\boldsymbol{q}$.

As shown in (Ou et al., 2025), by introducing a noise schedule $\gamma(t) = \frac{1}{t}$,[3] the random order autoregressive sampling of an MDM $\boldsymbol{\pi}_\theta$ can be viewed as a CTMC with the rate matrix $Q^\theta = (Q_t^\theta)_{t \in [0,1]}$ such that for $\boldsymbol{x} \neq \boldsymbol{y} \in \overline{\mathcal{V}}^D$,

$$Q_t^\theta(\boldsymbol{x}, \boldsymbol{y}) = \gamma(t)\boldsymbol{\pi}_\theta(\boldsymbol{x})_{d,n}, \text{ if } \boldsymbol{x}^d = \mathsf{M} \text{ and } \boldsymbol{y} = \boldsymbol{x}^{d \leftarrow n},$$

and 0 if otherwise, where $\boldsymbol{x}^{d \leftarrow n}$ means the sequence obtained by replacing the $d$-th position of $\boldsymbol{x}$ by $n$. The diagonal terms of $Q_t^\theta$ can be computed as

$$Q_t^\theta(\boldsymbol{x}, \boldsymbol{x}) = -\sum_{\boldsymbol{y} \neq \boldsymbol{x}} Q_t^\theta(\boldsymbol{x}, \boldsymbol{y}) = -\sum_{d:\boldsymbol{x}^d = \mathsf{M}} \sum_n Q_t^\theta(\boldsymbol{x}, \boldsymbol{x}^{d \leftarrow n})$$

$$= -\sum_{d:\boldsymbol{x}^d = \mathsf{M}} \sum_n \gamma(t)\boldsymbol{\pi}_\theta(\boldsymbol{x})_{d,n} = -\gamma(t) \cdot |\{d : \boldsymbol{x}^d = \mathsf{M}\}|. \tag{19}$$

Therefore, if $\mathbb{P}^\theta, \mathbb{P}^{\theta'}$ are the path measures of the sampling processes of two MDMs parameterized by $\theta$ and $\theta'$, respectively, then by (18), assuming that the jump from $\boldsymbol{\xi}_{t-}$ to $\boldsymbol{\xi}_t$ is at the $d(t)$-th position, we have

$$\log \frac{d\mathbb{P}^{\theta'}}{d\mathbb{P}^\theta}(\boldsymbol{\xi}) = \sum_{t:\boldsymbol{\xi}_{t-} \neq \boldsymbol{\xi}_t} \log \frac{\boldsymbol{\pi}_{\theta'}(\boldsymbol{\xi}_{t-})_{d(t),\xi_t^{d(t)}}}{\boldsymbol{\pi}_\theta(\boldsymbol{\xi}_{t-})_{d(t),\xi_t^{d(t)}}}, \quad \forall \boldsymbol{\xi} = (\boldsymbol{\xi}_t)_{t \in [0,1]}, \tag{20}$$

as the first term in (18) is always zero (both initial distributions are the point mass on the fully masked sequence), and the diagonal terms in the third term cancel out due to (19).

Moreover, as proved in Ou et al. (2025), the training of an MDM $\boldsymbol{\pi}_\theta$ given i.i.d. samples from the target distribution $p_{\text{data}}$ can be interpreted as minimizing the KL divergence between the target path measure $\mathbb{P}^*$ and the parameterized path measure $\mathbb{P}^\theta$, where $\mathbb{P}^*$ is defined as the path measure of the CTMC with rate matrix $Q_t^*(\boldsymbol{x}, \boldsymbol{x}^{d \leftarrow n}) = \gamma(t) \Pr_{\boldsymbol{X} \sim p_{\text{data}}}(X^d = n | \boldsymbol{X}^{\text{UM}} = \boldsymbol{x}^{\text{UM}})1_{x^d = \mathsf{M}}$, i.e., with the ground-truth conditional distribution. Moreover, one can derive

$$\text{KL}(\mathbb{P}^* \| \mathbb{P}^\theta) = \mathbb{E}_{p_{\text{data}}(\boldsymbol{x})} \mathbb{E}_{m \sim \text{Unif}\{1,...,D\}} \left[ \frac{D}{m} \mathbb{E}_{\mu_m(\widetilde{\boldsymbol{x}}|\boldsymbol{x})} \sum_{d:\widetilde{\boldsymbol{x}}^d = \mathsf{M}} -\log \boldsymbol{\pi}_\theta(\widetilde{\boldsymbol{x}})_{d,x^d} \right] + \text{const},$$

where const does not depend on $\theta$, and $\mu_m(\cdot|\boldsymbol{x})$ means to sample a uniformly random subset of $\{1, ..., D\}$ of size $m$ and mask the corresponding positions in $\boldsymbol{x}$. Note that this is exactly the denoising cross-entropy loss $\mathbb{E}_{p_{\text{data}}(\boldsymbol{x})} \mathcal{L}_\theta(\boldsymbol{x})$ as presented in (2). In other words, minimizing the KL divergence between sequence-level probabilities $p_*(\boldsymbol{\xi}_1)$ and $p_\theta(\boldsymbol{\xi}_1) \approx e^{-\mathcal{L}_\theta(\boldsymbol{\xi}_1)}$ in (11) can be interpreted as *precisely* minimizing the KL divergence between path-level probabilities $\mathbb{P}^*(\boldsymbol{\xi})$ and $\mathbb{P}^\theta(\boldsymbol{\xi})$.

---

[3]The choice of noise schedule is essentially not important for MDM. In fact, $\gamma$ can be any positive function with $\int_0^1 \gamma(t)dt = \infty$. Here, we follow the convention in most of the literature on MDM and choose this specific $\gamma$ such that the conditional distribution of $\boldsymbol{\xi}_t \in \overline{\mathcal{V}}^D$ given $\boldsymbol{\xi}_1 \in \mathcal{V}^D$ is obtained by independently masking each position in $\boldsymbol{\xi}_1$ with probability $1 - t$.

**Fine-tuning MDMs as a Stochastic Optimal Control Problem on Path Measures**    The task of fine-tuning a pretrained MDM can be viewed as a stochastic optimal control (SOC) problem on the space of path measures: given a pretrained MDM $\pi_{\mathrm{ref}}$ which generates a distribution $p_{\mathrm{ref}}$, we define its induced **reference path measure** as $\mathbb{P}^{\mathrm{ref}}$, with rate matrix $Q_t^{\mathrm{ref}}(\boldsymbol{x}, \boldsymbol{x}^{d\leftarrow n}) = \gamma(t)\boldsymbol{\pi}_{\mathrm{ref}}(\boldsymbol{x})_{d,n}1_{x^d=\mathsf{M}}$, and has terminal distribution $\mathbb{P}_1^{\mathrm{ref}} = p_{\mathrm{ref}}$. We aim at finding a target rate matrix $Q^*$ such that the associated target path measure $\mathbb{P}^*$ has a terminal distribution $p_*$ defined in the following way of tilting by reward:

$$p_*(\boldsymbol{x}) = \frac{1}{Z}p_{\mathrm{ref}}(\boldsymbol{x})\mathrm{e}^{r(\boldsymbol{x})/\alpha}, \ \boldsymbol{x} \in \mathcal{V}^D, \qquad \text{where } Z = \sum_{\boldsymbol{x}} p_{\mathrm{ref}}(\boldsymbol{x})\mathrm{e}^{r(\boldsymbol{x})/\alpha}.$$

This can be achieved by defining the target path measure $\mathbb{P}^*$ as

$$\mathbb{P}^*(\boldsymbol{\xi}) = \mathbb{P}^{\mathrm{ref}}(\boldsymbol{\xi}_{[0,1)}|\boldsymbol{\xi}_1)p_*(\boldsymbol{\xi}_1) = \mathbb{P}^{\mathrm{ref}}(\boldsymbol{\xi})\frac{p_*(\boldsymbol{\xi}_1)}{p_{\mathrm{ref}}(\boldsymbol{\xi}_1)} = \frac{1}{Z}\mathbb{P}^{\mathrm{ref}}(\boldsymbol{\xi})\mathrm{e}^{r(\boldsymbol{\xi}_1)/\alpha}, \ \forall \boldsymbol{\xi} = (\boldsymbol{\xi}_t)_{t\in[0,1]}. \tag{21}$$

We use a network $\boldsymbol{\pi}_\theta$ to parameterize the new rate matrix, initialized at $\boldsymbol{\pi}_{\mathrm{ref}}$. Given a current path measure $\mathbb{P}^\theta$ induced by a CTMC with rate matrix $Q_t^\theta(\boldsymbol{x}, \boldsymbol{x}^{d\leftarrow n}) = \gamma(t)\boldsymbol{\pi}_\theta(\boldsymbol{x})_{d,n}1_{x^d=\mathsf{M}}$, we can first derive the RN derivative between the path measures by (20):

$$\begin{aligned}
\log\frac{\mathrm{d}\mathbb{P}^*}{\mathrm{d}\mathbb{P}^\theta}(\boldsymbol{\xi}) &= \log\frac{\mathrm{d}\mathbb{P}^*}{\mathrm{d}\mathbb{P}^{\mathrm{ref}}}(\boldsymbol{\xi}) + \log\frac{\mathrm{d}\mathbb{P}^0}{\mathrm{d}\mathbb{P}^\theta}(\boldsymbol{\xi}) \\
&= \frac{r(\boldsymbol{\xi}_1)}{\alpha} - \log Z + \sum_{t:\boldsymbol{\xi}_{t-}\neq\boldsymbol{\xi}_t}\log\frac{Q_t^0}{Q_t^\theta}(\boldsymbol{\xi}_{t-},\boldsymbol{\xi}_t) + \int_0^1\sum_{y\neq\boldsymbol{\xi}_t}(Q_t^\theta - Q_t^0)(\boldsymbol{\xi}_t, y)\mathrm{d}t \\
&= \frac{r(\boldsymbol{\xi}_1)}{\alpha} + \sum_{t:\boldsymbol{\xi}_{t-}\neq\boldsymbol{\xi}_t}\log\frac{\boldsymbol{\pi}_{\mathrm{ref}}(\boldsymbol{\xi}_{t-})_{d(t),\xi_t^{d(t)}}}{\boldsymbol{\pi}_\theta(\boldsymbol{\xi}_{t-})_{d(t),\xi_t^{d(t)}}} - \log Z =: W^\theta(\boldsymbol{\xi}) - \log Z, \tag{22}
\end{aligned}$$

where we assume that the jump from $\boldsymbol{\xi}_{t-}$ to $\boldsymbol{\xi}_t$ is at the $d(t)$-th position. The idea of the weighted denoising cross-entropy (WDCE) loss is essentially to treat i.i.d. samples from the current policy $\mathbb{P}^\theta$ as weighted samples from $\mathbb{P}^*$, and minimizing the following loss:

$$\begin{aligned}
\mathrm{KL}(\mathbb{P}^*\|\mathbb{P}^\theta) + \text{const} &= \mathbb{E}_{p_*(\boldsymbol{x})}\mathcal{L}_\theta(\boldsymbol{x}) = \mathbb{E}_{\mathbb{P}^*(\boldsymbol{\xi})}\mathcal{L}_\theta(\boldsymbol{\xi}_1) \\
&= \mathbb{E}_{\mathbb{P}^v(\boldsymbol{\xi})}\frac{\mathrm{d}\mathbb{P}^*}{\mathrm{d}\mathbb{P}^v}(\boldsymbol{\xi})\mathcal{L}_\theta(\boldsymbol{\xi}_1) = \mathbb{E}_{\mathbb{P}^v(\boldsymbol{\xi})}\frac{1}{Z}\mathrm{e}^{W^v(\boldsymbol{\xi})}\mathcal{L}_\theta(\boldsymbol{\xi}_1),
\end{aligned}$$

where $\mathbb{P}^v$ is the path measure induced by a CTMC with rate matrix $Q^v$ where the network is parameterized by $v$ (e.g., the old parameters $\theta_{\mathrm{old}}$), whose parameters do not involve gradient calculation. For instance, we can set $v = \theta_{\mathrm{old}}$. Note that $Z = \mathbb{E}_{\mathbb{P}^v(\boldsymbol{\xi})}\mathrm{e}^{W^v(\boldsymbol{\xi})}$, which, if estimated via samples, is equivalent to doing softmax normalization on the logits $W^v(\boldsymbol{\xi})$ in the batch. Comparing with the WDCE loss (11) presented in Sec. 3.2, we conclude that they are essentially the same.

## B.2. Generalizing WDCE to Zero Temperature with Proximal Descent

Recall that our target distribution is (5), which is under a temperature $\alpha > 0$. We propose to generalize the WDCE loss (11) to incorporate the limiting case $\alpha \to 0$ from the viewpoint of **proximal descent** (Guo et al., 2025b).

The reward maximization problem (4) provides a variational characterization of the target distribution $p_*(\boldsymbol{o}|\boldsymbol{q})$. Suppose now we have a dLLM policy $\boldsymbol{\pi}_{\theta_{\mathrm{old}}}(\boldsymbol{o}|\boldsymbol{q})$ that outputs a distribution $p_{\theta_{\mathrm{old}}}(\boldsymbol{o}|\boldsymbol{q})$. We define the next target distribution $p_{\mathrm{tar}}(\boldsymbol{o}|\boldsymbol{q})$ as

$$p_{\mathrm{tar}}(\boldsymbol{o}|\boldsymbol{q}) = \underset{p_\theta(\boldsymbol{o}|\boldsymbol{q})}{\mathrm{argmax}}\left\{\mathbb{E}_{p_\theta(\boldsymbol{o}|\boldsymbol{q})}[r(\boldsymbol{q},\boldsymbol{o})] - \alpha\,\mathrm{KL}(p_\theta(\cdot|\boldsymbol{q})\|p_{\mathrm{ref}}(\cdot|\boldsymbol{q})) - \frac{1}{\eta'}\mathrm{KL}(p_\theta(\cdot|\boldsymbol{q})\|p_{\theta_{\mathrm{old}}}(\cdot|\boldsymbol{q}))\right\}, \tag{23}$$

where $\eta' > 0$ is the step size. Let $\eta = \frac{\eta'}{1+\eta'\alpha} \in \left(0, \frac{1}{\alpha}\right)$. It is easy to see that the solution is given by

$$\begin{aligned}
p_{\mathrm{tar}}(\boldsymbol{o}|\boldsymbol{q}) &\propto_{\boldsymbol{o}} p_{\theta_{\mathrm{old}}}(\boldsymbol{o}|\boldsymbol{q})^{1-\eta\alpha}p_{\mathrm{ref}}(\boldsymbol{o}|\boldsymbol{q})^{\eta\alpha}\mathrm{e}^{\eta r(\boldsymbol{q},\boldsymbol{o})}, \tag{24} \\
&\propto_{\boldsymbol{o}} p_{\theta_{\mathrm{old}}}(\boldsymbol{o}|\boldsymbol{q})^{1-\eta\alpha}p_*(\boldsymbol{o}|\boldsymbol{q})^{\eta\alpha}.
\end{aligned}$$

In fact, the term inside the brackets in (23) is $-\frac{1}{\eta} \text{KL}(p_\theta(\cdot|\boldsymbol{q})\|p_{\text{tar}}(\cdot|\boldsymbol{q})) + \text{const}$. This means the next target distribution is a geometric interpolation between the current model distribution $p_{\theta_{\text{old}}}$ and the optimal distribution $p_*$, with $\eta > 0$ being a step size parameter. (24) is well-defined even when $\alpha = 0$, although in this case, the target distribution concentrates on the set of maximizers of $r(\boldsymbol{q}, \boldsymbol{o})$ (e.g., all correct question-response pairs) without regularization from the base model $p_{\text{ref}}(\boldsymbol{o}|\boldsymbol{q})$.

For $\alpha = 0$, $p_{\text{tar}}(\boldsymbol{o}|\boldsymbol{q}) \propto_{\boldsymbol{o}} p_{\theta_{\text{old}}}(\boldsymbol{o}|\boldsymbol{q})e^{\eta r(\boldsymbol{q},\boldsymbol{o})}$. We can similarly solve the distribution matching problem via the WDCE loss:

$$\text{KL}(p_{\text{tar}}(\cdot|\boldsymbol{q})\|p_\theta(\cdot|\boldsymbol{q})) = \mathbb{E}_{p_{\text{tar}}(\boldsymbol{o}|\boldsymbol{q})}[-\log p_\theta(\boldsymbol{o}|\boldsymbol{q})] + \text{const}$$

$$= \mathbb{E}_{p_v(\boldsymbol{o}|\boldsymbol{q})} \underbrace{\frac{p_{\text{tar}}(\boldsymbol{o}|\boldsymbol{q})}{p_v(\boldsymbol{o}|\boldsymbol{q})}}_{=:w(\boldsymbol{o}|\boldsymbol{q})}[-\log p_\theta(\boldsymbol{o}|\boldsymbol{q})] + \text{const}$$

$$\leq \mathbb{E}_{p_v(\boldsymbol{o}|\boldsymbol{q})} w(\boldsymbol{o}|\boldsymbol{q})\mathcal{L}_\theta(\boldsymbol{o}|\boldsymbol{q}) + \text{const},$$

where the importance weight $w(\boldsymbol{o}|\boldsymbol{q}) \propto_{\boldsymbol{o}} \exp\left(\eta r(\boldsymbol{q},\boldsymbol{o}) + \log \frac{p_{\theta_{\text{old}}}(\boldsymbol{o}|\boldsymbol{q})}{p_{\theta_v}(\boldsymbol{o}|\boldsymbol{q})}\right)$. For $v \leftarrow \theta_{\text{old}}$, the weight simplifies to the softmax of $\eta r(\boldsymbol{q},\boldsymbol{o})$ over all responses for the same prompt $\boldsymbol{q}$. The weight baseline subtraction tricks also apply here.

We remark that when picking $\alpha = 0$, through the proximal gradient descent formulation, DMPO becomes completely *forward-only*, as it eliminates the need for estimating the sequence log probability ratio of the form $\log \frac{p_{\text{ref}}(\boldsymbol{o}|\boldsymbol{q})}{p_v(\boldsymbol{o}|\boldsymbol{q})}$, making it the best option to incorporate fast dLLM inference techniques for RL training speed-up. However, in this case, we can no longer guarantee the diversity in the target optimal distribution, and thus, we save this direction for future investigation.

### B.3. Insights for Weight Baselines: Approximate Variance Reduction

We first recall a classical equality in statistics regarding the **score function**: if $p_\theta(x)$ is a probability density or probability mass function parameterized by a continuous parameter $\theta$, then under certain weak regularity conditions, we have $\mathbb{E}_{p_\theta(x)} \nabla_\theta \log p_\theta(x) = 0$.

Therefore,

$$0 = \mathbb{E}_{p_\theta(\boldsymbol{o}|\boldsymbol{q})} \nabla_\theta \log p_\theta(\boldsymbol{o}|\boldsymbol{q}) = \nabla_\theta \mathbb{E}_{p_{\bar{\theta}}(\boldsymbol{o}|\boldsymbol{q})} \log p_\theta(\boldsymbol{o}|\boldsymbol{q})$$

$$= \nabla_\theta \mathbb{E}_{\boldsymbol{\sigma}} \mathbb{E}_{p_{\bar{\theta}}(\boldsymbol{o}|\boldsymbol{q};\boldsymbol{\sigma})} \log p_\theta(\boldsymbol{o}|\boldsymbol{q})$$

$$= \nabla_\theta \mathbb{E}_{\boldsymbol{\sigma}} \mathbb{E}_{p_v(\boldsymbol{o}|\boldsymbol{q};\boldsymbol{\sigma})} \frac{p_{\bar{\theta}}(\boldsymbol{o}|\boldsymbol{q};\boldsymbol{\sigma})}{p_v(\boldsymbol{o}|\boldsymbol{q};\boldsymbol{\sigma})} \log p_\theta(\boldsymbol{o}|\boldsymbol{q}).$$

Combined with (9), we can see that subtracting $\frac{p_{\bar{\theta}}(\boldsymbol{o}|\boldsymbol{q};\boldsymbol{\sigma})}{p_v(\boldsymbol{o}|\boldsymbol{q};\boldsymbol{\sigma})}$ from the weight does not change the gradient of the CE loss, i.e.,

$$\nabla_\theta \text{KL}(p_*(\cdot|q)\|p_\theta(\cdot|q)) = \nabla_\theta \mathbb{E}_{\boldsymbol{\sigma}} \mathbb{E}_{p_v(\boldsymbol{o}|\boldsymbol{q};\boldsymbol{\sigma})} \left(\frac{p_*(\boldsymbol{o}|\boldsymbol{q};\boldsymbol{\sigma})}{p_v(\boldsymbol{o}|\boldsymbol{q};\boldsymbol{\sigma})} - \lambda \frac{p_{\bar{\theta}}(\boldsymbol{o}|\boldsymbol{q};\boldsymbol{\sigma})}{p_v(\boldsymbol{o}|\boldsymbol{q};\boldsymbol{\sigma})}\right)[-\log p_\theta(\boldsymbol{o}|\boldsymbol{q})], \ \forall \lambda \in \mathbb{R}.$$

Theoretically, there is an optimal choice of $\lambda$ that minimizes the variance. The natural choice of $\lambda = 1$ means implicitly matching the probability $p_\theta(\boldsymbol{o}|\boldsymbol{q};\boldsymbol{\sigma})$ to fit $p_*(\boldsymbol{o}|\boldsymbol{q};\boldsymbol{\sigma})$, which corresponds to our model weight baseline (15). When the frequency for sampling buffer $F$ is small, we can assume $p_\theta(\boldsymbol{o}|\boldsymbol{q};\boldsymbol{\sigma})$ does not deviate too much from $p_v(\boldsymbol{o}|\boldsymbol{q};\boldsymbol{\sigma})$, thus this ratio should be close to 1, which corresponds to our group weight baseline (13). Finally, as we actually use the negative ELBO $\mathcal{L}_\theta(\boldsymbol{o}|\boldsymbol{q})$ instead of $-\log p_\theta(\boldsymbol{o}|\boldsymbol{q})$ in computing the loss, the variance reduction only holds *approximately*.

### B.4. Proofs for the Weighted Direct Discriminative Optimization Objective

For notational simplicity, we ignore the conditional dependence on $\boldsymbol{q}$. Write

$$\mathcal{F}(p_\theta) = -\mathbb{E}_{p_*} \log \frac{p_\theta}{p_\theta + p_v} - \mathbb{E}_{p_v} \log \frac{p_v}{p_\theta + p_v}.$$

For any fixed $\boldsymbol{o}$, consider the function

$$p_\theta(\boldsymbol{o}) \mapsto -p_*(\boldsymbol{o}) \log \frac{p_\theta(\boldsymbol{o})}{p_\theta(\boldsymbol{o}) + p_v(\boldsymbol{o})} - p_v(\boldsymbol{o}) \log \frac{p_v(\boldsymbol{o})}{p_\theta(\boldsymbol{o}) + p_v(\boldsymbol{o})}.$$

The derivative with respect to $p_\theta(\boldsymbol{o})$ is $-\frac{p_*(\boldsymbol{o})}{p_\theta(\boldsymbol{o})} + \frac{p_*(\boldsymbol{o})+p_v(\boldsymbol{o})}{p_\theta(\boldsymbol{o})+p_v(\boldsymbol{o})}$, which is $> 0$ if $p_\theta(\boldsymbol{o}) > p_*(\boldsymbol{o})$ and $< 0$ if $p_\theta(\boldsymbol{o}) < p_*(\boldsymbol{o})$. Therefore, this function is minimized at $p_\theta(\boldsymbol{o}) \leftarrow p_*(\boldsymbol{o})$, which completes the proof.

# C. Details of Experiments and Further Results

## C.1. Introduction of Datasets and Rewards used

To ensure a fair comparison, we use the same datasets and training rewards as d1 (Zhao et al., 2025a). For a self-contained presentation, we list the datasets and the rewards below.

**GSM8K.** GSM8k (Cobbe et al., 2021) is a mathematical reasoning dataset featuring multi-step grade school math problems. We conduct fine-tuning on the train split and evaluate on the test split.[4]

The reward is decomposed as follows:

1. *XML Structure Reward*: $+0.125$ for each correctly placed opening and closing tag (`<reasoning>`, `</reasoning>`, `<answer>`, `</answer>`) and $-0.001$ for each extra token after the closing tag `</answer>`.

2. *Soft Format Reward*: $+0.5$ for responses matching the pattern `<reasoning>...</reasoning><answer>...</answer>`.

3. *Strict Format Reward*: $+0.5$ for matching the specified format precisely with correct line breaks.

4. *Integer Answer Reward*: $+0.5$ if the retrieved answer parses as an integer.

5. *Correctness Reward*: $+2$ if the returned answer equals the ground truth exactly.

**MATH500.** MATH500 (Lightman et al., 2023) is a mathematical reasoning dataset, as well as a curated collection of 500 high-school-level problems sampled from the MATH (Hendrycks et al., 2021) dataset. We conduct fine-tuning on the train split and evaluate on the test split.[5]

The reward comprises

1. *Format Reward*: 1 when answer tags are present and \boxed appears inside them; $0.75$ when the tags are present but \boxed is absent; $0.50$ when the tags are missing but \boxed is present; $0.25$ when neither the tags nor \boxed appear.

2. *Correctness Reward*: $+2$ when the correct answer is enclosed in \boxed{}.

**Countdown.** Countdown (Pan et al., 2025) is a planning task that requires solving a combinatorial arithmetic challenge, which is to form a target number using basic arithmetic operations with a provided set of 3 numbers, where each number can only be used once. We train on the training split of the dataset from the TinyZero project (Pan et al., 2025), restricting to instances that use only three numbers, and evaluate on 256 synthetically generated countdown questions with three numbers.

The reward checks if an arithmetic expression constructed from given numbers reaches a target value. More specifically, it is 1 when the equation equals the target and uses exactly the available numbers, 0.1 when the equation uses the right numbers but does not reach the target, and 0 if otherwise.

**Sudoku.** Sudoku is a planning task that requires solving $4 \times 4$ Sudoku puzzles, which demand constraint satisfaction and logical elimination to correctly fill the grid. We use the training dataset from `https://github.com/Black-Phoen ix/4x4-Sudoku-Dataset`, in particular, the subset containing one million unique puzzles, which was synthetically generated using code from Arel (2025). For evaluation purposes, we randomly generate 256 Sudoku puzzles using this generator. The reward equals the fraction of originally blank cells that the model fills correctly.

## C.2. Training Hyperparameters and Evaluation

We choose the training hyperparameters following Zhao et al. (2025a) for a fair comparison. We also use the Transformer Reinforcement Learning library (TRL, von Werra et al. (2020)) to implement DMPO. During training, we also employed the same Low-Rank Adaptation (LoRA, Hu et al. (2022)) with a rank of $r = 128$ and scaling factor $\alpha = 64$. For all tasks, the training was conducted on 8 NVIDIA H100 or H200 GPUs with the hyperparameters described below.

---

[4]`https://huggingface.co/datasets/openai/gsm8k`
[5]`https://huggingface.co/datasets/ankner/math-500`

We use a maximum generation length 256 tokens, a batch size of 8 per GPU, and gradient accumulation steps of 2, and 16 generated rollouts per prompt. We optimized the model using the AdamW optimizer (Loshchilov & Hutter, 2019) with parameters $\beta_1 = 0.9, \beta_2 = 0.99$, weight decay of 0.1, learning rate of $3 \times 10^{-6}$, and gradient clipping at 0.2. For each clean sequence, we sampled 4 partially masked tokens to compute the WDCE/WDDO loss. For rollouts generation during training, we use a semi-autoregressive random order sampler (with temperature 0) and fast-dLLM (with temperature 0.2) with a block size of 32 to generate diverse responses, which is the recommended practice for using LLaDA series models as is described in Nie et al. (2025b). We train $4,000$ steps (number of gradient updates) for GSM8K and MATH500, Countdown, and Sudoku, respectively.

In Tab. 1, DMPO-LLaDA uses the group weight baseline (13) on GSM8K, MATH500, and Sudoku, and the individual weight baseline (14) on Countdown. DMPO-LLaDA-SFT and DMPO-LLaDA-1.5 adopt the individual weight baseline (14) in all cases.

For the reproduction of the d1 results, we follow the guidelines listed in Zhao et al. (2025a) and first perform SFT on s1k (Muennighoff et al., 2025) before applying diffu-GRPO. We use the recommended hyperparameter setups and train for up to $13,000$ iterations on each dataset before evaluating the results.

For computational efficiency, we use Flash Attention 2 (Dao, 2024) and 4-bit quantization. All experiments on DMPO share these hyperparameters. The main result reported in Tab. 1 used the group weight baseline defined in (13). The ablation study in Fig. 4 also follows the same set of hyperparameters above, except for using different choices of weight baselines.

For the evaluation of LLaDA-series checkpoints, we consider three different generation lengths: 128, 256, and 512. We correspondingly use 128, 256, and 512 steps for generation. For the LLaDA series of models, such as LLaDA-Instruct, LLaDA-1.5, d1-LLaDA, and our own DMPO-LLaDA, we employ the semi-autoregressive sampler with a block size of 32, a greedy decoding scheme with a temperature of 0, and the top-$k$ remasking scheme to achieve the best inference results. For the Dream model, we report generation lengths 256 and 512, and employ the recommended practice with temperature 0.95 and the top-$k$ remasking scheme.

## C.3. Further Experimental Results

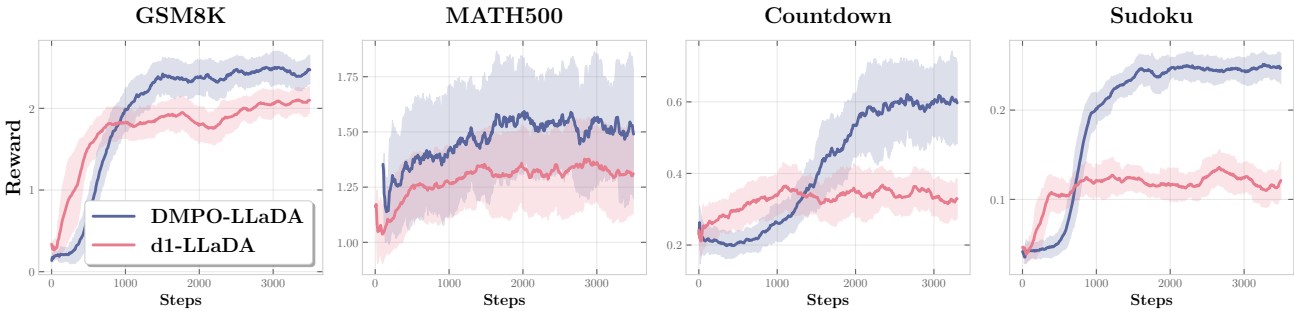

*Figure 6.* Reward dynamics during training. DMPO consistently produces higher rewards than d1.

**DMPO consistently achieves higher rewards**   In Fig. 6, we present the reward dynamics of DMPO across training steps and compare with that of d1. DMPO consistently achieves higher reward values after an initial warm-up phase and ultimately discovers responses with higher rewards than d1, possibly because it continuously explores the reward distribution landscape throughout training. In the first $1,000$ steps, DMPO often produces lower reward values than d1, potentially due to the lack of an SFT phase before RL scaling. Moreover, we observe that the performance of DMPO does not saturate after $4,000$ gradient steps, suggesting its greater potential than GRPO-type algorithms.

**Ablation studies on the hyperparameter dependence**   We provide an ablation study on two of the main hyperparameters in Alg. 1, namely the number of rollouts $N$ and the frequency for sampling buffer $F$, in Figs. 7 and 8, respectively. For each run shown in Fig. 7, we train for 6 hours using 8 NVIDIA H200 GPUs. For each run shown in Fig. 8, we train for 8 hours with 8 NVIDIA H200 GPUs. We only vary the resampling buffer frequency $F$ and the number of rollouts sampled per prompt $N$, while fixing other hyperparameters, such as the total effective batch size, to maintain a fair comparison.

For the number of rollouts per question $N$, we observe that a larger number of $N$ does not necessarily lead to longer training

time, even with the same number of steps, due to the parallelism of the generation process, since we kept the total batch size fixed while varying the hyperparameter $N$. The algorithm is robust across various values of $N$ ranging from $4$ to $32$ thanks to the mechanism for inserting negative gradients.

For the buffer sampling frequency $F$, we observe that it significantly affects training speed. The figure clearly demonstrates the advantage of DMPO due to its *off-policy* nature, whereas a purely on-policy realization of WDCE loss (with $F = 1$) is not only extremely slow but also does not show a significant boost in per-step reward gains. The figure also underscores the unique benefit of WDCE being a *forward* loss: given the generated rollouts and their weights, one can train using the simple forward process via random masking. Our algorithm is robust to choices of $F$ up to $24$, whereas an even larger $F$ may cause slight instability later in training when the reward is high.

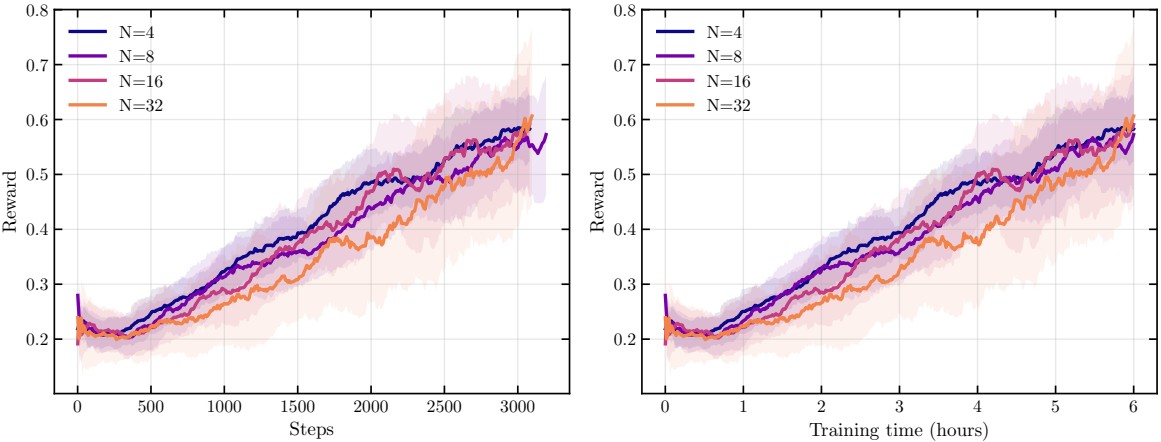

*Figure 7.* Ablation study of the number of rollouts per prompt $N$ on Countdown dataset under the same training time and compute. The performance is robust to this hyperparameter.

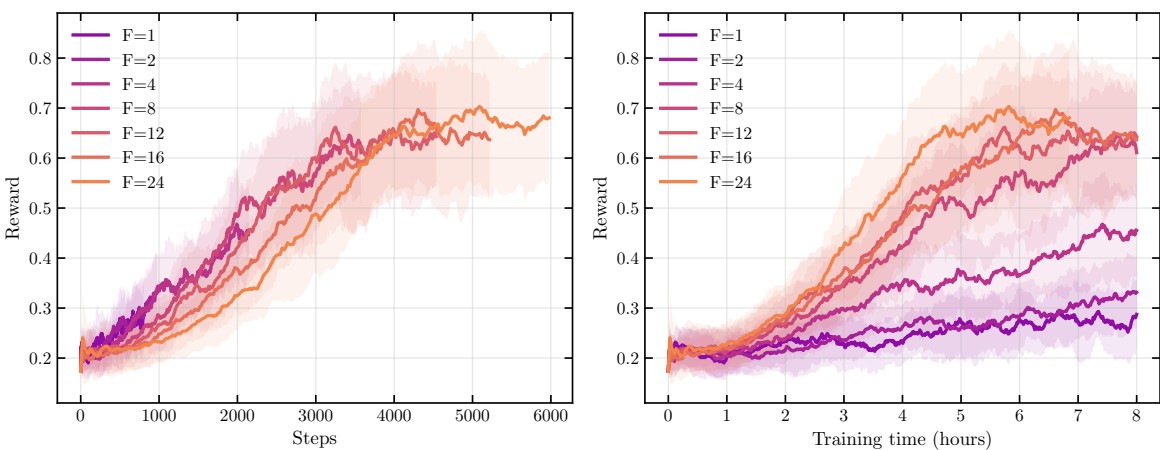

*Figure 8.* Ablation study of the resampling frequency $F$ on Countdown dataset. A larger $F$ is generally more time-efficient though may cause instability when the reward is high.

**Visualizing rollout entropy of DMPO**  In Fig. 9, we compare the reward and entropy of the generated rollouts during training for both the relative-entropy-based (diffu-GRPO) RL algorithm and the cross-entropy-based (DMPO) RL algorithm. Here, in both experiments, we fix $N = 16$ and $F = 8$ and evaluate the entropy of generated samples every 10 generations. The evaluation of entropy is as follows: we use random-order autoregressive generation with block length 32, and at the $d$-th step of unmasking (where $d$ ranges from 1 to $D = |\boldsymbol{o}|$), we compute the entropy of the predicted logits at the $d$-th position, and take average of all the $D$ entropy values as the final value of sequential entropy. From the figure, the trend of consistently higher sample entropy for WDCE loss than for diffu-GRPO agrees with our expectation that cross-entropy-based methods are less prone to mode-seeking and maintain a higher level of diversity throughout training.

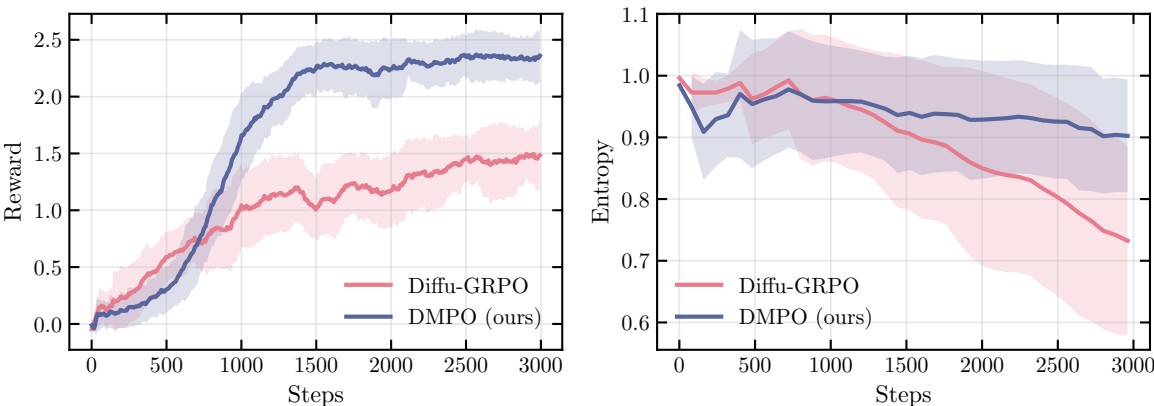

*Figure 9.* Reward (left) and entropy (right) of generated rollouts during training for both diffu-GRPO (Zhao et al., 2025a) and DMPO.

## C.4. Example Outputs of the Model after Fine-tuning

We present two example outputs of the DMPO-LLaDA model in the following.

**Model:** DMPO-LLaDA
**Dataset:** MATH500
**Generation Sequence Length:** $512$
**Question:** The product of $3t^2 + 5t + a$ and $4t^2 + bt - 2$ is $12t^4 + 26t^3 - 8t^2 - 16t + 6$. What is $a + b$?

To find the values of $a$ and $b$ such that the product of $3t^2 + 5t + a$ and $4t^2 + bt - 2$ equals $12t^4 + 26t^3 - 8t^2 - 16t + 6$, we need to expand the left-hand side and compare coefficients with the right-hand side.
First, expand the product:
$$(3t^2 + 5t + a)(4t^2 + bt - 2)$$

Using the distributive property (FOIL method for binomials):

$$3t^2 \cdot 4t^2 + 3t^2 \cdot bt + 3t^2 \cdot (-2) + 5t \cdot 4t^2 + 5t \cdot bt + 5t \cdot (-2) + a \cdot 4t^2 + a \cdot bt + a \cdot (-2)$$

Simplify each term:
$$12t^4 + 3bt^3 - 6t^2 + 20t^3 + 5bt^2 - 10t + 4at^2 + abt - 2a$$

Combine like terms:
$$12t^4 + (3b + 20)t^3 + (-6 + 5b + 4a)t^2 + (-10 + ab)t - 2a$$

We need this to equal:
$$12t^4 + 26t^3 - 8t^2 - 16t + 6$$

By comparing coefficients, we get the following system of equations:

$$3b + 20 = 26 \implies 3b = 6 \implies b = 2$$

$$-6 + 5(2) + 4a = -8 \implies 4 + 4a = -8 \implies 4a = -12 \implies a = -3$$

Thus, $a = -3$ and $b = 2$.
$$a + b = -3 + 2 = -1$$

</reasoning><answer> $\boxed{-1}$ </answer>

**Model:** DMPO-LLaDA
**Dataset:** GSM8K
**Generation Sequence Length:** 512
**Question:** Katarina has 5 less cookies than Max has. Max has 12 more cookies than the Cookie Monster, and Summer has 23 more cookies than Max. If Katarina has 68 cookies, how many cookies do they have in total?

To determine the total number of cookies Katarina, Max, and Summer have, we need to find out how many cookies each person has and then sum them up.
1. We know that Katarina has 68 cookies.
2. Katarina has 5 less cookies than Max.
3. Max has 12 more cookies than the Cookie Monster.
4. Summer has 23 more cookies than Max.
Let's denote the number of cookies each person has as follows:
- $K$ for Katarina's cookies
- $M$ for Max's cookies
- $C$ for the Cookie Monster's cookies
- $S$ for Summer's cookies
From the problem, we have:
- $K = 68$
- $K = M - 5$
- $M = C + 12$
- $S = M + 23$
First, we substitute the known value of $K$ into the equations:
- $68 = M - 5$
- $M = C + 12$
- $S = M + 23$
First, solve for $M$:
$$68 = M - 5$$
$$M = 68 + 5$$
$$M = 73$$

Next, solve for $C$:
$$73 = C + 12$$
$$C = 73 - 12$$
$$C = 61$$

Finally, solve for $S$:
$$73 = S - 23$$
$$S = 73 + 23$$
$$S = 96$$

Now, we have the number of cookies each person has:
- Katarina: 68 cookies
- Max: 73 cookies
- The Cookie Monster: 61 cookies
- Summer: 96 cookies
To find the total number of cookies, we sum these amounts:
$$68 + 73 + 61 + 96 = 298$$

Thus, the total number of cookies is $\boxed{298}$.
</reasoning><answer> $\boxed{298}$ </answer>

