# OpenReview forum: "Enhancing Reasoning for Diffusion LLMs via Distribution Matching Policy Optimization"
_ICML.cc/2026/Conference — ICML 2026 spotlight_

### Official Review · Reviewer_RP4b · 2026-03-10

**Soundness:** 3
**Presentation:** 3
**Significance:** 3
**Originality:** 3
**Overall Recommendation:** 5
**Confidence:** 3

**Summary:**

This manuscript proposes Distribution Matching Policy Optimization (DMPO), a novel reinforcement learning fine-tuning method tailored for diffusion Large Language Models (dLLMs). Shifting from the standard reward maximization paradigm (e.g., PPO, GRPO), DMPO targets matching a reward-tilted policy distribution using cross-entropy optimization. The authors implement this via a Weighted Denoising Cross-Entropy (WDCE) loss and introduce weight baseline subtraction techniques to stabilize training under small batch sizes.

**Compliance With Llm Reviewing Policy:**

Affirmed.

**Final Justification:**

After considering both the paper and the authors’ rebuttal, I am more positive about this work. The paper presents a novel and well-motivated distribution-matching perspective for RL fine-tuning of diffusion LLMs, and the empirical results are strong across multiple reasoning benchmarks.
My main initial concerns were the baseline choice, the treatment of stale-data covariate shift, and several presentation/practical issues. The rebuttal addressed these points well. In particular, the authors clarified the rationale for using different baselines, provided a convincing explanation for why DMPO avoids the instability typical of standard off-policy methods under stale data, and addressed the practical overhead concern of the model-weight baseline. The planned revisions on presentation also help.
While some limitations remain, such as validation beyond the LLaDA family and a more direct study of the ELBO approximation gap, these do not weaken the main contribution. Overall, the rebuttal resolved my main concerns, so I am increasing my score from 4 to 5.

**Key Questions For Authors:**

•	What is the principled justification for using the Group weight baseline for most tasks but switching to the Individual baseline for Countdown and the post-trained models?
•	How do you theoretically or empirically account for covariate shift when the buffer sampling frequency $F$ is large, given that the behavior policy becomes increasingly stale?
•	Have you measured the empirical gap between the true sequence log probability and the ELBO surrogate during active RL finetuning?

**Limitations:**

yes

**Strengths And Weaknesses:**

Claims And Evidence
The authors claim DMPO improves reasoning capabilities in dLLMs and enables efficient, off-policy training. The primary evidence is empirical: DMPO outperforms base models and GRPO-adapted baselines (like d1 and cGRPO) across four reasoning benchmarks (GSM8K, MATH500, Countdown, Sudoku). The authors also provide training dynamics curves as evidence of sample efficiency and stability when using stale data.
Methods And Evaluation Criteria
	Strengths: The formulation of RL for dLLMs as a forward KL distribution matching problem is mathematically elegant and uniquely suited to the diffusion framework. The use of importance sampling to render the WDCE loss off-policy allows for rollout reuse in a replay buffer, which addresses a major compute bottleneck inherent to on-policy algorithms.
	Weaknesses: The selection of the weight baseline function appears ad-hoc. The authors propose Group, Individual, and Model baselines, but rely on the Group baseline for GSM8K, MATH500, and Sudoku, while inexplicably switching to the Individual baseline for the Countdown task and for all post-trained models (LLaDA-SFT and LLaDA-1.5). This lack of methodological consistency undermines the robustness of the evaluation criteria.
Theoretical Claims
The theoretical foundation relies heavily on approximating intractable sequence log probabilities using the negative Evidence Lower Bound (ELBO). The authors justify this via Stochastic Optimal Control (SOC), demonstrating that minimizing sequence-level KL divergence with ELBO mathematically equates to matching exact path-level probabilities.
Experimental Designs Or Analyses
	Strengths: The experiments rigorously benchmark against the latest dLLM-specific RL algorithms (d1, cGRPO) using established reasoning datasets. The ablation studies on buffer sampling frequency $F$ and the number of rollouts $N$ provide highly useful practical insights for reproducibility.
	Weaknesses: The empirical validation relies exclusively on the LLaDA model family (LLaDA-Instruct, LLaDA-1.5), limiting the ability to claim broad architectural generalizability. Furthermore, the claim of off-policy stability with highly stale data e.g.,F=24 completely ignores the theoretical covariate shift. As the model policy diverges from the behavior policy over $F$ gradient steps, the importance sampling weights should degrade, yet this divergence is not quantified.
Supplementary Material
The supplementary material is robust and essential, containing critical mathematical proofs for the SOC framework (Appendix B) and comprehensive experimental hyperparameter details (Appendix C).
Relation To Broader Scientific Literature
The manuscript effectively situates itself within the rapid evolution of RL for LLMs (PPO, GRPO) and their recent adaptations to discrete diffusion spaces. It adequately references concurrent distribution-matching RL methods like FlowRL and MDNS.
Essential References Not Discussed
The literature review is thorough and appropriately covers concurrent works up to the submission deadline. No essential references are visibly absent.
Other Strengths And Weaknesses
A minor weakness is the visual clarity of the rollout entropy graphs (Figure 9); the confidence intervals heavily overlap in the first half of training, making the statistical significance of the claims regarding diversity preservation difficult to assess visually. A practical strength is the seamless integration of DMPO with fast decoding algorithms e.g.,fast-dLLM to heavily accelerate training.
Other Comments Or Suggestions
The SOC proof is central to justifying the ELBO surrogate and should be briefly summarized in the main text (Section 3.2) rather than purely relegated to Appendix B. Additionally, the computational memory overhead for calculating the "Model" weight baseline—which requires estimating the negative ELBO under the current model policy —should be explicitly discussed, as it impacts deployment at scale.

---

> ### Author Rebuttal · Authors · 2026-03-31
>
> We thank the reviewer for the constructive feedback.
>
> ---
>
> **W1 & Q1: Justification for weight baseline choices.**
>
> We investigated both baselines across all tasks. The Individual baseline serves as a robust **unified default**, performing well universally and particularly strongly on Countdown and post-trained models whose distributions differ from base LLaDA-Instruct. The Group baseline yields slightly better absolute performance on reasoning-heavy tasks (GSM8K, MATH500, Sudoku) by reducing intra-batch variance. We will include a cross-task comparison in revision. (See also our response to reviewer p7jo W2.)
>
> **W2: Limited base model scale and architecture.**
>
> Due to space limit, please refer to our response to reviewer p7jo (W1).
>
> **W3 & Q2: Covariate shift with highly stale data.**
>
> We appreciate the reviewer for raising this important theoretical concern. However, we respectfully clarify that covariate shift in the importance weights is **entirely avoided** in DMPO. As we explicitly detailed in Sec. 4 (under the paragraph "DMPO exhibits stable training despite highly stale data"), WDCE is inherently **off-policy**, and the importance sampling weights depend strictly on $\pi_*/\pi_{\rm old}$. The current model policy $\pi_\theta$ is **absent from** the importance weight. Therefore, even if $\pi_\theta$ is updated over $F$ steps and diverges from $\pi_{\rm old}$, no exploding $\pi_\theta/\pi_{\rm old}$ ratios occur. In stark contrast, for standard policy gradient methods, importance weights depend on the ratio $\pi_\theta/\pi_{\rm old}$, which inevitably exhibit instability as the active policy $\pi_\theta$ is updated over $F$ steps. Empirically, Figs. 5 and 8 confirm this stability: DMPO maintains monotonic reward improvement with buffer frequencies up to $F=24$, without performance degradation.
>
> **W4: Visual clarity of the rollout entropy graphs (Fig. 9).**
>
> We appreciate this feedback on the visual presentation. The overlapping confidence intervals in the first half of training accurately reflect the high variance and active exploration typical of the early RL phase, where the policy has not yet significantly diverged from the initial state. However, as the policies converge in the latter half of training, the gap in rollout entropy becomes distinct and statistically significant, supporting our core claim that DMPO better preserves generation diversity in the converged model. We will improve the clarity in revision.
>
> **W5: SOC proof should be summarized in the main text.**
>
> We agree with the reviewer and will add a concise summary of the SOC proof in the main text in revision.
>
> **W6: Computational memory overhead of the model weight baseline.**
>
> In our implementation, calculating the model baseline (Eq. 15) introduces **zero additional computational or memory overhead**, as we approximate it by the ELBO. The ELBO under the current model policy is inherently evaluated as the standard forward pass during the WDCE loss computation. To construct the weight baseline, we simply **detach and reuse** this computed scalar value. No secondary forward passes, additional memory allocations, or separate gradient graphs are required.
>
> **Q3: Gap between sequence log probability and ELBO.**
>
> We did not measure the exact gap, as computing the true sequence log-likelihood in dLLMs is intractable as it requires marginalization over all possible unmasking orders. While $K$-sample bounds (e.g., IWAE [1]) could theoretically approximate this gap, their bias and variance scale as $O(1/K)$ with a constant governed by the relative variance of importance weights [2], which is prohibitively large in the combinatorial trajectory space of dLLMs.
>
> We acknowledge this is a valid concern. However, we note that the WDCE loss (Eq. 11) closely resembles the standard denoising cross-entropy (DCE) used in pretraining and SFT, differing only at the importance weights. Since the pretrained model was successfully trained using this same ELBO surrogate, and our use of KL regularization with the pretrained model (temperature $\alpha$) constrains the policy from diverging too far, we expect the ELBO quality to remain reliable throughout RL finetuning. Moreover, as shown in [3], the optimal solution of the WDCE objective is itself a valid MDM, suggesting that the optimization does not fundamentally degrade the model's MDM structure. Empirically, the stable and monotonic reward curves (Fig. 6) and consistent accuracy gains across all tasks support this. Quantitatively tracking ELBO quality during RL finetuning is an important future direction.
>
> [1] Burda et al. Importance Weighted Autoencoders. arXiv:1509.00519.
>
> [2] Nowozin. Debiasing Evidence Approximations: On Importance-weighted Autoencoders and Jackknife Variational Inference. ICLR 2018.
>
> [3] Zhu et al. MDNS: Masked Diffusion Neural Sampler via Stochastic Optimal Control. NeurIPS 2025.
>
> ---
> We thank the reviewer again for the detailed feedback and hope our response clarifies the concerns.

---

> > ### Author Rebuttal · Reviewer_RP4b · 2026-04-02
> >
> > I really appreciate the authors' professional rebuttal. It exactly resolved all my concerns. I would like to adjust my score accordingly (from 4 to 5).

---

> > > ### Author Response · Authors · 2026-04-07
> > >
> > > We sincerely thank the reviewer for the positive reassessment and for raising the score. We are glad that our rebuttal has fully resolved the concerns. The constructive feedback has been very helpful in strengthening the paper. Thank you again for your time and engagement in the discussion!

---

### Official Review · Reviewer_QBKZ · 2026-03-13

**Soundness:** 2
**Presentation:** 3
**Significance:** 2
**Originality:** 2
**Overall Recommendation:** 4
**Confidence:** 3

**Summary:**

RL for diffusion LLM have several difficulties, since (1) the generation needs multiple call of the model (2) sequence likelihood is intractable where we only have an ELBO surrogate since there are so many ways to decode a sentence. Prior GRPO methods for dLLMs have tried to efficiently approximate the likelihood such as using mean-field approximation as done in diffu-GRPO, coupled-GRPO. But this treats the tokens are conditionally independent of each other, resulting in noisy RL signal and variances in training.

This paper proposes to do directly distribution matching, where the optimal policy of entropy regularized RL optimization can be written as proportional to the reference policy times exponential term of the reward divide by a temperature parameter constant. It uses order-specific token factorization to compute unnormalized importance weights. and then fit the dllm to the samples with the weighted weights computed ealier. They use forward KL for distribution matching.

**Compliance With Llm Reviewing Policy:**

Affirmed.

**Final Justification:**

The authors have addressed my questions and concerns. i will remain my positive score.

**Key Questions For Authors:**

1. Forward KL encourages mode covering. While this increases diversity, it may also assign probability mass to suboptimal trajectories. Do you have any discussion on the trade-off between forward and reverse kl for distribution matching for dLLM?
2. Is same order used to generate the rollout is used in the denoising loss computation? is the order randomly decided or based on some heuristic such as high-confidence unmasking?

**Limitations:**

1. The gain seems limited as compared to several recent policy-gradient based methods for dLLM such as SPG: https://arxiv.org/pdf/2510.09541. Especially on countdown and sudoku.
2. the WDCE loss also approximates per-token log prob by sampling masking pattern and use one forward pass to get the masked token's logits, this also assume conditional independence among the masked tokens? which was done in diffugrpo, coupledgrpo. Would this also create variance in the importance weight calculation?

**Strengths And Weaknesses:**

1. This paper is theoretically motivated and creates a new angle on how to optimizing reward-based reasoning for dLLM, where prior works have mostly focused on policy-gradient based methods.
2. This method has advantage in being off-policy where prior rollouts can be reused for better efficiency then purely on-policy learning like in diffu-grpo.
3. this methods shows strong reasoning improvements over baselines and is tested across various sequence length for comprehensiveness
4. DMPO consistently maintains higher sample entropy than baseline methods, which maintains diversity for RL training.

---

> ### Author Rebuttal · Authors · 2026-03-31
>
> We thank the reviewer for the thoughtful and constructive feedback. We are glad to hear that the reviewer found our work theoretically motivated and shows strong reasoning improvements.
>
> ---
>
> **Q1: Trade-off between forward and reverse KL**
>
> This is an excellent observation. Forward KL is indeed mode-covering, which improves diversity but risks assigning probability mass to suboptimal trajectories (low-reward reasoning paths). However, we explicitly mitigate this theoretical trade-off through our negative gradient insertion and weight baseline subtraction mechanisms (Sec. 3.3). By doing so, trajectories that fall below the expected weight threshold receive a negative signal, thus penalizing the suboptimal modes that forward KL might otherwise over-explore, effectively balancing diversity (mode-covering) with high-reward exploitation. Empirically, Fig. 9 provides supporting evidence: DMPO consistently maintains higher sample entropy than diffu-GRPO throughout training while achieving higher rewards, confirming that forward KL with our weight baseline successfully preserves diversity without sacrificing performance.
>
> **Q2: Rollout order and denoising loss computation**
>
> We would like to clarify that the rollout generation order and the denoising loss computation are **completely decoupled** in our framework.
>
> - For the WDCE loss computation: The generation order is not used. Since WDCE is a **forward-only** objective, we simply treat the generated rollouts as clean data and apply forward masking. Specifically, we sample the number of masked tokens $m\sim\text{Unif}\\{1,...,|\mathbf{o}|\\}$ and randomly mask $m$ positions uniformly across the sequence.
> - For rollout generation: In our main results (Tab. 1), all rollouts are generated using Fast-dLLM (Wu et al., 2025), which employs an approximate KV-cache mechanism with semi-autoregressive decoding at a block size of 32. Regardless of the internal decoding strategy used during generation, DMPO discards the generation order entirely before computing the WDCE loss. Only the final clean samples and their rewards are retained.
>
> **L1: Comparison with SPG**
>
> We acknowledge SPG's strong empirical results, particularly on Countdown and Sudoku. Importantly, DMPO and SPG explore **orthogonal directions**: SPG optimizes within the policy gradient framework (improving PG via EUBO), while DMPO introduces an entirely different paradigm: distribution matching grounded in SOC. These directions are complementary rather than competing, and their contributions could potentially be combined in future work.
>
> DMPO's distinctive value lies in its **off-policy, forward-only** nature. SPG and all GRPO-based methods are on-policy, requiring continuous real-time rollouts, which is computationally expensive for dLLMs. DMPO's WDCE objective enables massive stale-data reuse (up to $F=24$ in Fig. 8) and seamless integration with fast inference techniques, achieving the same reward level as d1 with only 31% of the training budget (1.8 vs 5.78 PF-days, Fig. 5).
>
> We also note that the evaluation setups differ: SPG evaluates Sudoku under a **3-shot** setting while we use **0-shot**, making direct comparison on Sudoku not straightforward. On Countdown, DMPO actually **outperforms** SPG at seq len 256 and 512 (80.86% / 82.81% vs. 70.7% / 70.3%). Furthermore, during the rebuttal period, DMPO achieved **60.25% accuracy on Sudoku (seq len 256, 0-shot)** with extended training (up from 24.56%), suggesting that the performance ceiling is far from reached. We will include a thorough discussion of SPG in our revised related work.
>
> **L2: Conditional independence assumption in WDCE and variance in importance weights**
>
> We clarify that DMPO avoids the mean-field approximation used by d1 and cGRPO. The critical distinction lies in how masked tokens are handled:
>
> - **d1**: Fully masks **all** response tokens and uses a **single** forward pass. Each token is predicted with **zero context from other response tokens**, making this a true mean-field approximation that assumes conditional independence, introducing significant variance.
> - **DMPO (WDCE)**: Randomly masks only **a subset** of $m$ out of $|\mathbf{o}|$ tokens. Each masked token is predicted with **rich context from the unmasked portion of the response**, which inherently captures inter-token dependencies within each forward pass. Moreover, by averaging over multiple different masking patterns (sampling different $m$ values as in Eq. 2), the ELBO aggregates information across diverse conditioning contexts, effectively capturing the joint distribution without relying on mean-field assumptions.
>
> Therefore, the variance issue caused by conditional independence in d1/cGRPO does not apply to our setting.
>
> ---
>
> We thank the reviewer again for the detailed feedback and hope our responses clarify the concerns. We are happy to provide any additional information as needed.

---

> > ### Author Rebuttal · Reviewer_QBKZ · 2026-04-03
> >
> > I appreciate the detailed explanation. It has resolved my concerns and I remain my positive score.

---

> > > ### Author Response · Authors · 2026-04-07
> > >
> > > We sincerely thank the reviewer for confirming that our rebuttal has fully resolved the concerns. During the discussion period, we have also conducted substantial additional experiments that further strengthen the paper, including applying DMPO to a Dream-7B-Instruct and comparing against additional concurrent baselines (wd1, GDPO). These results are detailed in our second response to Reviewer 5QFG and will be incorporated into the revised manuscript.
> > >
> > > Given these additions and improvements to the paper, we would be grateful if the reviewer could consider adjusting the score to further support the paper. Thank you again for your time and engagement in the discussion!

---

### Official Review · Reviewer_5QFG · 2026-03-13

**Soundness:** 3
**Presentation:** 3
**Significance:** 3
**Originality:** 2
**Overall Recommendation:** 3
**Confidence:** 2

**Summary:**

This paper studies RL-style post-training for diffusion language models (dLLMs) on reasoning tasks. Instead of directly maximizing reward, it formulates training as matching a reward-tilted target distribution and introduces DMPO, a practical optimization framework based on weighted denoising cross-entropy and related weighted objectives. The method is evaluated on LLaDA-based models across GSM8K, MATH500, Sudoku, and Countdown, with additional analyses on weight baselines, stale-data training, and training efficiency.

**Compliance With Llm Reviewing Policy:**

Affirmed.

**Final Justification:**

The new experiments address my empirical concerns regarding generalization and statistical validation. I raise my soundness score accordingly, while noting that the core framework remains closely derived from MDNS and the conceptual novelty is moderate.

**Key Questions For Authors:**

refer to the weakness

**Limitations:**

yes

**Strengths And Weaknesses:**

Strengths

- RL optimization for diffusion language models is still relatively underexplored, and adapting reasoning-oriented policy optimization to dLLMs is a meaningful research direction.

- Recasting the objective as reward-tilted distribution matching is conceptually clean, and the derivation of practical weighted objectives is reasonably well aligned with the structure of masked diffusion models.

- The paper reports consistent gains over the base dLLM and RL baselines on four reasoning benchmarks.

Weaknesses

- The originality is limited relative to the closest prior work. A substantial part of the technical framework, especially the distribution-matching perspective and the WDCE-style practical objective, is explicitly built on the MDNS line of work. The main contribution here is therefore closer to adapting and extending that framework to dLLM reasoning post-training than to introducing a fundamentally new optimization principle. This remains a central concern.

- The experiments are centered on the LLaDA family and only four reasoning/planning benchmarks. While the reported gains are encouraging, this setup is not yet sufficient to establish that the method generalizes broadly across diffusion LLMs, tasks, or post-training settings. This limitation appears to remain in the current version.

- The paper includes useful ablations, but the results still do not completely disentangle whether the improvements come primarily from the distribution-matching objective itself, the weighting scheme, stale-data reuse, or other implementation choices.

- The paper mainly compares against d1 and cGRPO-style baselines. A sufficient comparison to recent and related diffusion-LM optimization methods would make the empirical gains more convincing.

- The experiments don't provide multi-seed variance or significance tests.

---

> ### Author Rebuttal · Authors · 2026-03-31
>
> We thank the reviewer for the detailed and constructive feedback.
>
> **W1: Novelty and relation to MDNS**
>
> We clarify our key distinctions from MDNS as follows. Almost all existing dLLM RL methods (d1, cGRPO) directly adapt the GRPO paradigm from AR-LLMs. In contrast, DMPO introduces a **fundamentally different paradigm**: shifting from reward-maximizing policy gradients to **distribution matching**.
>
> While we draw mathematical inspiration from MDNS, DMPO is far from a trivial adaptation. They tackle completely different problem settings and face entirely different optimization bottlenecks. Specifically:
>
> 1. **Core task**: MDNS trains neural samplers, whereas DMPO fine-tunes dLLMs.
> 2. **Optimization goal**: MDNS targets exact matching of a given unnormalized target distribution, whereas DMPO maximizes reasoning rewards while maintaining regularization with a pre-trained model.
> 3. **Batch size**: MDNS does not specifically address constrained settings, whereas DMPO operates in a highly constrained RL setting (e.g., 16 rollouts per prompt).
> 4. **Key innovation**: MDNS uses standard WDCE, whereas DMPO introduces weight baseline subtraction to prevent collapse under small batch sizes, and the WDDO objective (Sec. 3.4).
>
> **W2: Limited model scope and benchmarks**
>
> Our evaluation aligns with the standard settings established by d1 and cGRPO, where LLaDA-8B is the recognized testbed. We note that DMPO's core components, i.e., the WDCE loss and weight baseline subtraction, depend only on the general MDM framework shared by both LLaDA and DREAM. Neither the importance weight computation (Eq. 10) nor the forward noising process makes assumptions specific to LLaDA's architecture, suggesting strong potential for broader applicability. We will add a discussion on architecture generalizability in the revision. Regarding benchmarks, our four tasks (GSM8K, MATH500, Countdown, Sudoku) span both mathematical reasoning and constraint planning. Furthermore, Tab. 1 shows that DMPO delivers consistent gains when applied to base models with different post-training histories: LLaDA-SFT (CoT SFT) and LLaDA-1.5 (DPO), confirming its robustness across diverse starting points.
>
> **W3: Disentangling the improvements**
>
> Rather than a collection of independent tricks, the components of DMPO are mathematically unified. However, we have indeed provided empirical ablations in the paper that isolate these components:
>
> - **Isolating the distribution-matching objective from stale-data reuse**: In Fig. 8, we evaluated the model with a buffer sampling frequency of $F=1$, which completely removes "stale-data reuse". The results show that the model still achieves stable and significant reward gains, albeit requiring more wall-clock time. This explicitly proves that *the distribution-matching objective itself is the fundamental driver of the reasoning improvements, while stale-data reuse strictly acts as an accelerator for computational efficiency*.
> - **Isolating the weighting scheme**: In Fig. 4, we explicitly ablated the weighting mechanism. The "WDCE (No Weight Baseline)" curve demonstrates that without weight baseline subtraction, the core objective fails to improve under the constrained small batch-size setting. This disentangles the weighting scheme, proving *it is a necessary bridging mechanism to make the distribution-matching objective viable for RL fine-tuning*.
>
> We will add a summary in the revision to explicitly connect these existing figures and clarify how each component uniquely contributes to the final performance.
>
> **W4: Comparison with recent baselines**
>
> All results in Tab. 1 are reproduced under our unified evaluation protocol, ensuring a strictly fair comparison. Moreover, concurrent works often adopt different evaluation setups (e.g., different test splits, few-shot settings, generation length conventions), making cross-paper numerical comparison potentially misleading. While some concurrent works such as wd1 have released training code, trained checkpoints are not publicly available, making it difficult to conduct a controlled comparison under identical evaluation conditions. We will expand the related work discussion in the revision to thoroughly contextualize DMPO relative to these concurrent methods.
>
> Additionally, during the rebuttal period, we conducted further experiments showing that DMPO achieves **60.25% accuracy on Sudoku (sequence length 256)** with extended training, a substantial improvement over the 24.56% reported in the submission, highlighting significant room for further improvement.
>
> **W5: Multi-seed variance and significance tests**
>
> We agree that multi-seed variance would strengthen the evaluation. We note that multi-seed RL fine-tuning at the 8B scale is computationally demanding, and current dLLM RL baselines (d1, cGRPO) similarly report single-run results. Nevertheless, the consistent improvement margins across four distinct tasks with different reward structures provide evidence of robustness.

---

> > ### Author Rebuttal · Reviewer_5QFG · 2026-04-04
> >
> > The authors provide reasonable clarifications on technical distinctions and existing ablations, but the core concerns regarding generalization beyond LLaDA, broader baseline comparisons, and multi-seed statistical validation remain unaddressed; I therefore maintain my current score.

---

> > > ### Author Response · Authors · 2026-04-06
> > >
> > > We thank the reviewer for the continued engagement. We are glad that our clarifications on W1 and W3 were found reasonable. Below, we provide substantial new experimental results conducted during the rebuttal and discussion period to directly address the three remaining concerns.
> > >
> > > **W2: Generalization beyond LLaDA: New experiments on Dream-7B**
> > >
> > > We have applied DMPO to Dream-7B-Instruct, which is another major family of diffusion LLMs apart from LLaDA. We also applied d1 (diffu-GRPO) to Dream to serve as an RL baseline on this architecture. We report multi-seed results (mean ± std across 3 seeds for statistical validation) below:
> > >
> > > | | GSM8K | | MATH500 | | Countdown | | Sudoku | |
> > > |---|---|---|---|---|---|---|---|---|
> > > | **Model / Seq Len** | **256** | **512** | **256** | **512** | **256** | **512** | **256** | **512** |
> > > | Dream-Instruct (7B) | 73.39 | 76.65 | 36.60 | 36.40 | 28.52 | 27.34 | 16.41 | 11.77 |
> > > | d1-Dream† | 80.52 | 82.41 | 41.20 | 45.60 | 29.46 | 36.83 | 21.78 | 23.34 |
> > > | DMPO-Dream (Ours) | 84.03±0.21 | 84.76±0.23 | 47.40±0.69 | 47.20±0.60 | 54.43±1.76 | 56.51±1.19 | 41.04±0.93 | 45.55±2.55 |
> > >
> > > †Reproduced using released codebase.
> > >
> > > DMPO delivers large and consistent improvements on Dream-Instruct (7B) across all four benchmarks, substantially outperforming d1-Dream at sequence length 256: +3.51% on GSM8K, +6.20% on MATH500, +24.97% on Countdown, and +19.26% on Sudoku. These gains are comparable to or even larger than those observed on LLaDA, confirming that DMPO generalizes across different dLLM architectures. Notably, DMPO requires no architecture-specific modifications: the same WDCE loss and weight baseline subtraction apply directly to Dream, as we stated in our initial response.
> > >
> > > **W4 & W5: Broader baseline comparisons and multi-seed variance**
> > >
> > > We provide new results comparing DMPO against concurrent methods wd1 and GDPO on LLaDA, alongside multi-seed statistics (mean ± std across 3 seeds for statistical validation).
> > >
> > > | | GSM8K | | MATH500 | | Countdown | | Sudoku | |
> > > |---|---|---|---|---|---|---|---|---|
> > > | **Model / Seq Len** | **256** | **512** | **256** | **512** | **256** | **512** | **256** | **512** |
> > > | LLaDA-Instruct (8B) | 79.76 | 83.62 | 35.00 | 38.80 | 14.45 | 14.84 | 6.10 | 7.37 |
> > > | d1-LLaDA† | 81.40 | 84.38 | 36.60 | 40.80 | 26.56 | 30.47 | 11.04 | 8.69 |
> > > | cGRPO-LLaDA† | 81.73 | 84.23 | 32.80 | 38.40 | 42.58 | 37.11 | 24.17 | 21.97 |
> > > | wd1-LLaDA† | 80.80 | 82.31 | 37.60 | 39.80 | 49.22 | 47.17 | 22.04 | 24.65 |
> > > | GDPO-LLaDA* | 81.20 | 82.26 | 38.00 | 38.20 | 67.19 | 66.41 | 24.17 | 25.10 |
> > > | DMPO-LLaDA (Ours) | 82.23±0.16 | 84.94±0.27 | 37.54±1.29 | 41.40±1.64 | 81.12±0.23 | 81.90±0.81 | 63.80±4.79 | 56.67±1.15 |
> > >
> > > †Reproduced using released codebase. *Results taken from the original paper.
> > >
> > > DMPO achieves the best performance on nearly all benchmarks and setups and outperforms concurrent baseline methods like wd1 and GDPO. The multi-seed standard deviations are also seen universally small on most benchmarks (e.g., GSM8K std < 0.3%, Countdown std < 1%), demonstrating strong reproducibility and technical robustness of the method.
> > >
> > > ----
> > > We will incorporate all the above results into the revised version of the paper to enhance the paper and further demonstrate the robustness and effectiveness of DMPO across architectures and random seeds. We hope the new experiments on Dream-7B, broader baselines comparison, and multi-seed results adequately address the reviewer's remaining concerns, and we kindly ask the reviewer to reconsider the rating in light of these additions that addressed your concerns.

---

### Official Review · Reviewer_p7jo · 2026-03-15

**Soundness:** 4
**Presentation:** 4
**Significance:** 3
**Originality:** 3
**Overall Recommendation:** 5
**Confidence:** 3

**Summary:**

This paper proposes Distribution Matching Policy Optimization (DMPO), an RL fine-tuning method for diffusion large language models (dLLMs). The intuition is that instead of reward maximizatino, DMPO targets matching the full reward-tilted distribution via forward KL minimization. One important benefit of DMPO is the improved efficiency obtained through the foward loss. Experiments on four reasoning benchmarks using LLaDA-8B show DMPO consistently outperforms the prior SOTA d1 method with especially large gains on planning tasks and significant compute savings.

**Compliance With Llm Reviewing Policy:**

Affirmed.

**Final Justification:**

Author thoroughly addressed my concerns and therefore I increased my score to 5.

**Key Questions For Authors:**

See weakness section. Recommended experiments are listed as well.

**Limitations:**

yes

**Strengths And Weaknesses:**

## Strength
S1. The idea of shifting reward maximization to distribution matching is well-motivated. The stochastic optimal control provides a clean theoretical grounding. Additionally, the equivalence between sequence-level ELBO matching and path-level KL minimization is a satisfying result that justifies the ELBO approximation used in practice.

S2. The empirical gains are substantial, especially on planning tasks, and the compute efficiency is remarkable. It can have significant impact on improving the current diffusion language model landscape.

## Weakness
W1. All experiments only use a single base model family (LLaDA-8B). How the method performs regarding different model size should be acknowledged and discussed. Additionally, how DMPO performs on different dLLM architectures is worth investigating as well (e.g. on DREAM-7B).

W2. Additionally, the main results in Table 1 use different weight baselines for different tasks (group baseline for GSM8K/MATH500/Sudoku, individual baseline for Countdown). This inconsistency makes it difficult to assess the robustness of any single configuration. The ablation in Figure 4 is only on Sudoku; a cross-task ablation of baseline choices would clarify whether task-specific tuning of the baseline type is necessary in practice.

W3. The temperature parameter α is important for controlling the target distribution but its selection is not discussed. Similarly, the paper does not report how rollout generation is split between the ROAR sampler and Fast-dLLM in the main Table 1 results, making reproducibility harder.

---

> ### Author Rebuttal · Authors · 2026-03-31
>
> We sincerely thank the reviewer for recognizing the strong motivation of our distribution-matching approach, the clean theoretical grounding of SOC, and the substantial empirical gains DMPO achieves. We deeply appreciate the constructive suggestions, which have been instrumental in strengthening the robustness and clarity of our work.
>
> ---
>
> **W1: Base model scale and architectures.**
>
> We acknowledge that our current experiments focus on the LLaDA-8B family, which aligns with the evaluation settings of foundational dLLM RL works (e.g., d1, cGRPO). While scaling to different architectures (like DREAM-7B) or larger parameter counts requires significant computational resources beyond our current capacity, we will explicitly acknowledge this limitation and add a dedicated discussion on architecture generalizability in revision. Importantly, DMPO's core components, i.e., the WDCE loss and weight baseline subtraction, depend only on the general masked diffusion model (MDM) framework shared by both LLaDA and DREAM. Neither the importance weight computation (Eq. 10) nor the forward noising process makes assumptions specific to LLaDA's architecture, suggesting strong potential for broader applicability.
>
> **W2: Inconsistency of weight baselines and cross-task performance.**
>
> We thank the reviewer for raising this point and agree that a unified baseline is practically desirable. We clarify that we did investigate the cross-task performance of both baselines.
>
> Empirically, the Individual baseline provides robust and stable performance across all tasks, including Countdown where it significantly outperforms the Group baseline. Therefore, the Individual baseline can serve as a strong **unified default choice** with minimal performance trade-offs.
>
> However, on reasoning-heavy tasks (GSM8K, MATH500, Sudoku), we observed that the Group baseline yields slightly better absolute performance by more effectively reducing intra-batch variance. To report the absolute capability and performance ceiling of DMPO, we opted to present the Group baseline results for these tasks in Tab. 1. We acknowledge that achieving the absolute optimal performance currently benefits from task-specific baseline tuning. As requested, we will explicitly include this cross-task baseline comparison, discuss these trade-offs, and recommend the Individual baseline as a unified default in the revised appendix.
>
> **W3: Temperature $\alpha$ and rollout generation split.**
>
> We thank the reviewer for pointing this out and will include these details in the revision to ensure full reproducibility.
>
> For the temperature $\alpha$, we used $\alpha=0.04$ across all experiments. Mathematically, $\alpha$ acts as the KL regularization coefficient. To ensure a strictly fair comparison, we adopted this exact value from the official d1 codebase and applied it uniformly to DMPO and the baselines.
>
> For the main results in Tab. 1, rollouts were generated using Fast-dLLM. We acknowledge that the wording in App. C.2 may cause confusion, as it lists hyperparameters for both the semi-autoregressive random order sampler and Fast-dLLM together; the ROAR sampler is used separately in the throughput comparison (Fig. 5). We will revise the appendix to clearly disambiguate these two settings.
>
> ----
>
> We thank the reviewer again for the thorough evaluation and valuable feedback. We hope our responses address your concerns, and we are more than happy to provide any further information if needed.

---

> > ### Author Rebuttal · Reviewer_p7jo · 2026-04-01
> >
> > Thank you for the reply. I would maintain my score.

---

> > > ### Author Response · Authors · 2026-04-07
> > >
> > > We thank the reviewer for the time and effort in providing a thoughtful review and constructive comments.
> > >
> > > Regarding W1, we have further provided experimental results on Dream-7B-Instruct to address the concern on DMPO's performance on other dLLM architectures. We also applied d1 (diffu-GRPO) to Dream to serve as an RL baseline on this architecture. We report multi-seed results (mean ± std across 3 seeds for statistical validation) below:
> > >
> > > | | GSM8K | | MATH500 | | Countdown | | Sudoku | |
> > > |---|---|---|---|---|---|---|---|---|
> > > | **Model / Seq Len** | **256** | **512** | **256** | **512** | **256** | **512** | **256** | **512** |
> > > | Dream-Instruct (7B) | 73.39 | 76.65 | 36.60 | 36.40 | 28.52 | 27.34 | 16.41 | 11.77 |
> > > | d1-Dream† | 80.52 | 82.41 | 41.20 | 45.60 | 29.46 | 36.83 | 21.78 | 23.34 |
> > > | DMPO-Dream (Ours) | 84.03±0.21 | 84.76±0.23 | 47.40±0.69 | 47.20±0.60 | 54.43±1.76 | 56.51±1.19 | 41.04±0.93 | 45.55±2.55 |
> > >
> > > †Reproduced using released codebase.
> > >
> > > DMPO delivers large and consistent improvements on Dream-Instruct (7B) across all four benchmarks, substantially outperforming d1-Dream at sequence length 256: +3.51% on GSM8K, +6.20% on MATH500, +24.97% on Countdown, and +19.26% on Sudoku. These gains are comparable to or even larger than those observed on LLaDA, confirming that DMPO generalizes across different dLLM architectures. Notably, DMPO requires no architecture-specific modifications: the same WDCE loss and weight baseline subtraction apply directly to Dream.
> > >
> > > In addition, we have also compared DMPO against concurrent baselines wd1 and GDPO on LLaDA with multi-seed statistics. The detailed results can be found in our second response to Reviewer 5QFG, where DMPO achieves the best performance on nearly all benchmarks and setups. All of these results will be incorporated into the revised manuscript to enhance the paper's quality and conclusion.
> > >
> > > ---
> > >
> > > Given these additional experiments and improvements to the paper, we would be grateful if the reviewer could consider adjusting the score to further support the paper. Thank you again for your time and engagement in the discussion!

---

### Decision · Program_Chairs · 2026-04-30

**Decision:**

Accept (spotlight)

**Comment:**

The paper proposes a new methodology to fine-tune diffusion llm by matching the reward tilted distribution of the policy. Such method can work offline and it is thus practical, and it is theoretical well justified as recognized by there reviewers. Most recommend acceptance, and thus I am glad to endorse the paper